# Methane exchange at the peatland forest floor – automatic chamber system exposes the dynamics of small fluxes

Mika Korkiakoski[1], Juha-Pekka Tuovinen[1], Mika Aurela[1], Markku Koskinen[2], Kari Minkkinen[2], Paavo Ojanen[3], Timo Penttilä[3], Juuso Rainne[1], Tuomas Laurila[1], Annalea Lohila[1]

[1]Finnish Meteorological Institute, Atmospheric Composition Research, P.O. Box 503, FI-00101 Helsinki, Finland
[2]University of Helsinki, Department of Forest Sciences, P.O. Box 27, FI-00014 University of Helsinki, Finland
[3]Natural Resources Institute Finland, Viikinkaari 4, FI-00790 Helsinki, Finland

*Correspondence to*: Mika Korkiakoski (mika.korkiakoski@fmi.fi)

**Abstract.** We measured methane ($CH_4$) exchange rates with automatic chambers at the forest floor of a nutrient-rich drained peatland in 2011–2013. The fen, located in southern Finland, was drained for forestry in 1969 and the tree stand is now a mixture of Scots pine, Norway spruce, and pubescent birch. Our measurement system consisted of six transparent chambers and stainless steel frames, positioned on a number of different field and moss layer compositions. Gas concentrations were measured with an on-line cavity ring-down spectroscopy gas analyzer. Fluxes were calculated with both linear and exponential regression. The use of linear regression resulted in systematically smaller $CH_4$ fluxes by 10–45 % as compared to exponential regression. However, the use of exponential regression with small fluxes ($< 2.5$ µg $CH_4$ $m^{-2}$ $h^{-1}$) typically resulted in anomalously large absolute fluxes and high hour-to-hour deviations. Therefore, we recommend that fluxes are initially calculated with linear regression to determine the threshold for "low" fluxes and that higher fluxes are then recalculated using exponential regression. The exponential flux was clearly affected by the length of the fitting period when this period was $< 190$ s, but stabilised with longer periods. Thus, we also recommend the use of a fitting period of several minutes to stabilise the results and decrease the flux detection limit. There were clear seasonal dynamics in the $CH_4$ flux: the forest floor acted as a $CH_4$ sink particularly from early summer until the end of the year, while in late winter the flux was very small and fluctuated around zero. However, the magnitude of fluxes was relatively small throughout the year, ranging mainly from $-130$ to $+100$ µg $CH_4$ $m^{-2}$ $h^{-1}$. $CH_4$ emission peaks were observed occasionally, mostly in summer during heavy rainfall events. Diurnal variation, showing a lower $CH_4$ uptake rate during the daytime, was observed in all of the chambers, mainly in the summer and late spring, particularly in dry conditions. It was attributed more to changes in wind speed than air or soil temperature, which suggest that physical rather than biological phenomena are responsible for the observed variation. The annual net $CH_4$ exchange varied from $-104\pm30$ to $-505\pm39$ mg $CH_4$ $m^{-2}$ $yr^{-1}$ among the six chambers, with an average of $-219$ mg $CH_4$ $m^{-2}$ $yr^{-1}$ over the two-year measurement period.

**Keywords:** methane uptake, methane emission, high resolution flux measurement, automatic chamber, peatland forest, drained peatland, linear regression, exponential regression

# 1 Introduction

Methane ($CH_4$) is one of the most important atmospheric greenhouse gases due to its capability to absorb thermal radiation and warm the climate (IPCC, 2014). One of the main sources of $CH_4$ globally are peatlands (e.g. Denman et al., 2007) where $CH_4$ is produced by the decomposition of organic matter in anaerobic conditions. Around 3 % (ca. 4 000 000 $km^2$) of the

5 Earth's land surface is covered by peatlands (Clarke and Rieley, 2010) and the majority of these are located in the boreal region (Fischlin et al., 2007). About one third (104 000 $km^2$) of European mire and peat resources are located in Finland (Montanarella et al., 2006) and more than half (55 000 $km^2$) of this area has been drained for forestry (Päivänen and Hånell, 2012).

Methane can be both produced and consumed in soil so that the net $CH_4$ flux depends on the rate of $CH_4$ production in anoxic soil layers and on the rate of $CH_4$ oxidation in the oxic soil layers. In peatlands, the thickness and depth of $CH_4$ producing and

10 oxidizing layers are largely determined by the water table (Bubier and Moore, 1994), which controls the vertical distribution of oxygen in the soil profile. $CH_4$ is produced under anaerobic conditions by microbes known as methanogens. The production rate is dependent on the availability of organic substrates at low redox potential (Eh) values, and is controlled by soil temperature and pH (Dunfield et al., 1993; Wang et al., 1993; Segers, 1998; Kotsyurbenko et al., 2004). On the other hand, oxidation of $CH_4$ occurs in the oxic soil layer closer to the surface and potentially also in the moss layer (Larmola et al., 2010).

Like the production rate, the oxidation rate is controlled by soil temperature and pH (Dunfield et al., 1993; Scheutz and Kjeldsen, 2004; Boeckx and Van Cleemput, 1996), but also many other factors affect oxidation processes, such as soil water content, soil texture, nutrients and $CH_4$ and oxygen concentration (Boeckx and Van Cleemput, 1996; Ridgwell et al., 1999; Scheutz and Kjeldsen, 2004). In addition to the direct control of production and oxidation rates, there are other phenomena which may affect the observed net $CH_4$ flux above the soil surface, including lateral $CH_4$ transport in the soil (Christophersen

and Kjeldsen, 2001) and subsurface storage (Hutchinson et al., 2000).

In environments with low soil $CH_4$ production, such as upland forest soils, grasslands and tundra, uptake of atmospheric $CH_4$ by the methanotrophic microbes dominates (Dutaur and Verchot, 2007). This is also what commonly happens after the drainage of peatlands, which results in water level drawdown and increased oxic layer thickness. Thereby, $CH_4$ production is decreased and the fraction of oxidized $CH_4$ increased (e.g. Moore and Knowles, 1989; Roulet et al., 1992). Consequently, the $CH_4$

oxidation rate in the aerated surface soil and mosses typically exceeds $CH_4$ production that occurs deeper in the soil, thus turning in particular well-drained peatlands into net $CH_4$ sinks (Martikainen et al., 1995; Minkkinen et al., 2007; Ojanen et al., 2010; Lohila et al., 2011). However, poorly drained sites may remain to act as $CH_4$ sources (Ojanen et al., 2010). In addition, the drainage ditches even at well-drained sites typically continue to emit $CH_4$ at rates similar to pristine boreal peatlands (Minkkinen et al., 1997; Minkkinen and Laine, 2006; Luan and Wu, 2015).

Closed chambers are commonly used in the measurement of greenhouse gas exchange between the forest floor and the atmosphere (e.g. Denmead, 2008; Forbrich et al., 2010, Koskinen et al., 2014). Unlike the eddy covariance (EC) method, which is more suitable for measuring fluxes at the ecosystem level, the chamber method permits the investigation of small-scale processes, such as the gas exchange of different microtopographic surfaces, and enables the quantification of spatial variation

(Keller et al., 1990; Singh et al., 1997). However, there are various details related to the chamber design and the deployment of this measurement technique in practice that may have a significant impact on the flux that is estimated from the observed concentration change in the chamber headspace. For example, the flux estimate seems to depend on the dimensions of the chamber (Pihlatie et al., 2013). In addition, chambers should include a fan to evenly distribute the air in the chamber headspace

(Pumpanen et al., 2004; Christiansen et al., 2011) although the rotational speed of the fan should be kept low to avoid excessive turbulence (Koskinen et al., 2014). A major source of uncertainty is the impact of the chamber itself on the gas concentration gradient in the soil (Healy et al., 1996; Hutchinson et al., 2000; Conen and Smith, 2000; Davidson et al., 2002; Livingston et al., 2005) and in the boundary layer just above it. The concentration gradient is critical as it drives the soil-atmosphere gas exchange and thus any aerodynamic disturbance may impact the observed flux.

The gradient between the soil and the air inside the chamber changes when the gas concentration inside the chamber changes during the measurement. This changes the flux, which makes the concentration change non-linear in time. However, non-linearity of the concentration during the chamber closure may also result from chamber leaking. For example, Pirk et al. (2016) demonstrated that the degree of convex curvature in the increasing methane concentration correlated positively with wind speed outside the chamber. Furthermore, in the case of soil acting as a methane sink, the methane consumption by soil

methanotrophs obeys the first-order reaction kinetics which should lead to curvilinear concentration dynamics in the chamber (e.g. Sabrekov et al., 2016). However, the different processes responsible for the curvature in the concentration time series may be difficult to separate from each other (Kutzbach et al., 2007).

There are many studies that have recognized that the use of linear regression in flux calculation can cause significant underestimation of the flux (e.g. Healy et al., 1996; Hutchinson et al., 2000; Nakano et al., 2004; Livingston et al., 2005; 2006;

Kutzbach et al., 2007; Kroon et al., 2008; Pedersen et al., 2010; Pihlatie et al., 2013). However, many studies have used linear regression (e.g. Laine et al., 2006; Alm et al., 2007; Jones et al., 2011; Bergier et al., 2013; Fassbinder et al., 2013), because under field conditions it is more robust to random measurement errors than non-linear methods. Moreover, the use of linear regression is preferred when comparing measurement sites as it is not as sensitive as non-linear models to small differences in soil properties (Venterea et al., 2009). The selection of the optimal fitting method is important as it can be a large source of

uncertainty in flux calculations (Levy et al., 2011; Venterea et al., 2013).

Although several studies have examined the different fitting methods for calculating fluxes from chamber data, there exist only a few papers exploring the dynamics of $CH_4$ flux data that mainly consist of small uptake fluxes and are measured with automatic chambers coupled to a high-resolution gas analyzer (e.g. Savage et al., 2014; Ueyama et al., 2015). In this study, we measured the $CH_4$ flux between a forest floor and the atmosphere continuously throughout 2 years at a boreal nutrient-rich

forestry-drained peatland site with typically small $CH_4$ exchange rates. We used six automatic soil chambers and a cavity ring-down spectroscopy analyzer, which allowed us to measure with a relatively high sampling rate during each chamber closure and to perform multiple daily measurements with each chamber. Our particular aims with this set-up were to determine:

1. What is the optimal fitting method for calculating the $CH_4$ flux?
2. How large are the diurnal, seasonal and inter-annual variations in the $CH_4$ flux?

3. What is the annual $CH_4$ balance of the study site?

## 2 Materials and methods

### 2.1 Site description

The measurements were made in southern Finland at Lettosuo (60°38' N, 23°57' E) (Fig. S1a), which is a nutrient-rich peatland
forest that was drained in 1969 and fertilized with phosphorus and potassium soon after. The open ditches, located in approximately 45 m intervals (Fig. S1b), were originally about 1 m deep but have since been partly filled with new vegetation. Before drainage, the tree stand was dominated by Scots pine (*Pinus sylvestris*) with some pubescent birch (*Betula pubescens*). After drainage, the stand has developed to a mixture of Scots pine and pubescent birch in the dominant canopy layer, with an understorey of Norway spruce (*Picea abies*) with some scattered small-sized pubescent birch. The stem volumes at the time
of this study equaled to 174, 46, and 28 $m^3$ $ha^{-1}$ for Scots pine, pubescent birch, and Norway spruce, respectively. The tree stand is quite dense, which results in irregular shading and consequently, patchy and variable ground vegetation layer. For example, herbs such as *Dryopteris carthusiana* and *Trientalis europaea*, and dwarf shrubs such as *Vaccinium myrtillus* are common in the ground vegetation (Bhuiyan et al., 2017). In addition, the moss layer is patchy and is dominated by *Pleurozium schreberi* and *Dicranum polysetum* with some *Sphagna (Sphagnum girgensohnii, Sphagnum angustifolium* and *Sphagnum*
*russowii)* appearing in moist patches.

CN ratio of the surface peat, sampled at four points located at a 20–40 m distance from the chamber plots, averaged 24 for the 0–20 cm layer (Table 1). The relatively low CN ratio is typical for fertile peatland forests and reflects the fen history of the site. The bulk density of these samples was 0.11 and 0.17 g $cm^{-3}$ for the 0–10 and 10–20 cm layers, respectively, while the average bulk density of the 0–20 cm layer below each chamber varied from 0.03 to 0.13 g $cm^{-3}$ (Koskinen et al., 2014). The
ash content of the peat varied from 3.4 to 6.5 %.

The vascular green area (VGA) was estimated for each chamber and vascular plant species every two weeks during the growing seasons 2011 and 2012 (Ojanen, unpublished data). This was done by estimating the number and dimensions of leaves within each chamber and calculating green area by species-specific regression models between leaf dimensions and green area. For *Vaccinium myrtillus*, also the surface area of the green stems was included into VGA. The coverages of the mosses was
estimated visually. The maximum VGA and the coverages for each chamber are shown in Table 2.

### 2.2 Flux measurement system and ancillary measurements

The automatic chamber measurement system is the same as used for $CO_2$ exchange by Koskinen et al. (2014). The $CO_2$ flux measurements started in autumn 2010, and the $CH_4$ analyzer was added to the system in March 2011. Here we report the $CH_4$ fluxes measured since then until April 2013. Forest floor gas exchange, including the tree roots, was monitored using six
transparent soil chambers connected to an instrument cabin. The cabin was located at a distance of about 30 m from the 25.5

m tall EC mast (Koskinen et al., 2014) from which the wind speed above the canopy was measured. The locations of the chambers were selected to maximize the number of different ground vegetation compositions (Table 2) within a circle of ca. 15 m radius around the cabin.

The details of the chamber system can be found in Koskinen et al. (2014), and thus here we only describe the main features of the system. The size of the chamber boxes were 57 cm x 57 cm x 30 cm (length x width x height). We used a permanently installed steel collar (height 5 cm, inserted at a depth of 2 cm) below each chamber to minimize the disturbance to the soil, and to enhance the sealing between the soil and the chamber. There was a U-profile at the bottom of the chamber edges, insulated with a foam tape, to further improve the sealing. In winter, the whole chamber frame was raised above the snow level by placing one or two extension collars (height 16 cm) between the frame and soil.

A 24 V fan (Maglev KDE2408PTV1, Sunon Ltd, Kaohsiung, Taiwan) (size 8 cm x 8 cm) was used to mix the air inside the chamber headspace. The voltage of the fan was regulated to keep the mixing steady, but as low as possible (Koskinen et al., 2014). Sample gas was drawn from the chamber typically once an hour (with some exceptions explained below) at a flow rate of about 1 L min$^{-1}$, and returned back to the chamber from the gas analyzers. $CH_4$, $CO_2$ and water vapor concentrations were measured approximately every 4 s with a Picarro G1130 cavity ring-down spectroscopy gas analyzer (Picarro Inc., Santa Clara, CA, USA). The inlet and outlet gas tubes (FESTO Oy, Vantaa, Finland) were made of polyurethane and were 15 m in length and had an inner and outer diameter of 4 mm and 6 mm, respectively.

The tubes were flushed with ambient air just before the chamber was closed. When all the chambers were open, ambient air was sampled. The delay in the analyzer response caused by the long tubing was taken into account using a flagging system in the computer program collecting the data, which labelled each data point with the respective chamber number using a 20 s lag. However, as the flow rate varied slightly in time, some points were removed from the data before the flux calculation (Sect. 2.4).

Air and soil temperature data were collected every 10 s using Pt100 probes (PT4T, Nokeval Oy, Nokia, Finland) and Nokeval 680-loggers (Nokeval Oy, Nokia, Finland). One probe was located inside each chamber at a height of 30 cm and positioned next to the fan under a metal heat shield to prevent direct solar radiation from affecting the measurements. Furthermore, soil surface temperature was monitored inside each chamber just below the surface of the moss or litter layer. In addition, soil temperature probes were placed at depths of 2, 5, 10, 20 and 30 cm at one location near the chambers. Water table level (WTL) was monitored every hour at four different points at the site (TruTrack WT-HR data loggers, Intech Instruments Ltd, Auckland, New Zealand). The air pressure, precipitation and snow depth data were acquired from the nearby Finnish Meteorological Institute observatory at Jokioinen (~35 km northwest of Lettosuo).

**2.3 Meteorological conditions**

The climate at the site has both continental and maritime influences. The annual mean temperature and precipitation at the nearby weather station in 1981−2010 were 4.6 °C and 627 mm, respectively (Pirinen et al., 2012). During this study, the first measurement year (4/2011−3/2012) was significantly warmer (annual mean temperature 5.8 °C) than the second measurement

year (4/2012–3/2013) (1.4 °C) (Fig. 1). The first year was slightly warmer and the second year was significantly colder than the long-term mean recorded at the nearby weather station (4.6 °C). Both the summer (JJA) (17.6 °C) and winter (DJF) (2.7 °C) temperatures in 2011 were warmer than those of 2012 (12.1 °C and –2.3 °C). In particular, the beginning of summer 2012 was much colder than the same period in 2011.

Annual precipitation during the first (976 mm) and second (780 mm) measurement years was higher than the long-term mean (627 mm). Summertime precipitation was 9% higher in the first (309 mm) year as compared to the second (284 mm) year, while in winter the difference was 18 % (577 mm and 490 mm in the first and second winters, respectively). The first snow appeared on 5 December in 2011 and 25 October in 2012, and the first permanent snow was recorded on 7 January in 2012 and 28 November in 2012. In spring (MAM) 2011, the snow had melted by 13 April. For spring 2012, we do not know the

exact day of snowmelt due to missing data, although the snow had melted at latest by 4 April. From the temperature data we estimate that the snow cover disappeared sometime in mid-March.

WTL varied from –8 to –59 cm from the soil surface (negative sign denotes WTL below the surface) and was highest in the spring and late autumn (SON). The lowest (i.e. deepest) values were recorded at the end of summer. The average WTL in summer 2011 was –47.2±7.4 cm (±SD) and –49.1±7.1 in summer 2012. Occasional sudden increases in WTL were observed

after rainfall events and it usually took 1–2 weeks to reach the WTL observed prior to the event.

## 2.4 Flux calculation

During the study period, the chambers were operated with varying closure times ranging from 2 to 16 minutes. In 2011, 2 min closures were used with the exception of 6 min measurements made four times per day. After mid-March 2012, the minimum closure time was 6 min. Thus, each chamber was typically sampled once an hour, with the exception of summer 2012 (JJA)

when a longer closure time of 16 min was tested and each chamber was sampled every two hours. For the analysis of $CH_4$ exchange dynamics (Sects. 3.3–3.4), we used the fluxes calculated with a 6 min closure time (as justified in Sect. 3.2). In addition to removing 20 s from the start of the measurement due to lag caused by long tubing (Sect. 2.2), 18 s were discarded from the start of a measurement to ensure that the air inside the chamber was properly mixed. Dilution and spectral corrected $CH_4$ concentrations reported by Picarro G1130 were used to calculate the fluxes.

Two different regression types were fitted to the data: linear and exponential. The linear function describing the change in the concentration, $C$, as a function of time was:

$$C(t) = a_{lin} + b_{lin}t, \tag{1}$$

where $a_{lin}$ and $b_{lin}$ are parameters and $t$ is the time from the start of the closure. In this model, the slope $b_{lin}$ equals the concentration change in time.

The exponential function we fitted was:

$$C(t) = a_{exp} + b_{exp}\exp(c_{exp}t), \tag{2}$$

where $a_{exp}$, $b_{exp}$ and $c_{exp}$ are parameters. When differentiating Eq. (2) with respect to time and inspecting the moment when the chamber closes ($t$=0), it follows that the concentration change with time is the product of parameters $b_{exp}$ and $c_{exp}$. It is generally considered that this initial rate of concentration change best represents the flux at that time. However, when fitting the exponential function to the data using the least squares approach, the fitting frequently fails due to local minima. To overcome this and to avoid over-parameterisation, a Taylor power series expansion (Kutzbach et al., 2007) was fitted to the data to determine initial estimates of the parameters of the exponential regression.

The exponential regression should capture the flux better than the linear regression as it takes into account the change in the gradient between soil and chamber headspace during chamber closure, which is evident when diffusion flux is decreasing the concentration difference. However, exponential regression is very sensitive to possible disturbances to the data at the beginning of chamber closure. In our study, we attempted to minimize these disturbances by closing the chamber slowly and smoothly, which seemed to prevent pressure fluctuations related to chamber closing. For the analysis of CH$_4$ exchange dynamics (Sects 3.3–3.4), we used flux data that are based on a combination of linear and exponential fits: first all fluxes were calculated using the linear regression, and below and above a limit of 2.5 µg CH$_4$ m$^{-2}$ h$^{-1}$ the fluxes were calculated with the linear and exponential method, respectively (for justification see Sect. 3.1).

The CH$_4$ flux ($F$, µg CH$_4$ m$^{-2}$ h$^{-1}$) was calculated according to Eq. 3, which is based on the ideal gas law:

$$F = \left(\frac{dC(t)}{dt}\right)_{t=0} \frac{MPV}{RTA} 3600 \frac{s}{h}, \tag{3}$$

where $\left(\frac{dC(t)}{dt}\right)_{t=0}$ is the time derivative (ppm s$^{-1}$) of a linear ($b_{lin}$) or exponential ($b_{exp} \times c_{exp}$) regression at the beginning of the closure, $M$ is the molecular mass of CH$_4$ (16.042 g mol$^{-1}$), $P$ is air pressure (Pa), $R$ is the universal gas constant (8.31446 J mol$^{-1}$ K$^{-1}$), $T$ is the mean chamber headspace temperature during closure (K), and $V$ and $A$ are the volume (m$^3$) and the base area (m$^2$) of the chamber headspace, respectively. Here, a micrometeorological sign convention is used: a positive flux indicates a flux from the ecosystem to the atmosphere (emission) and a negative flux indicates a flux from the atmosphere into the ecosystem (uptake).

When estimating the volume of the chamber headspace, the height of the moss and snow surfaces was assumed to represent the interface between the soil and air. In other words, the pore space in the soil and snow was ignored from the headspace volume. The error caused by this in flux calculations was estimated to be only a few percent (Koskinen et al., 2014). To create a continuous data set of snow depth, the manual measurements carried out irregularly at the site were combined with those measured daily at the Jokioinen observatory. In addition to snow depth, the height of the chamber headspace was measured at the start and end of the growing season from 16 points inside each collar by gently placing the end of a tape measure on top of the surface mosses (Koskinen et al., 2014). The height of the chamber headspace between these manual measurements was determined with linear interpolation.

All the calculations and analyses were made with the Python programming language (Python Software Foundation, version 2.7, https://www.python.org) using the following libraries: NumPy (http://www.numpy.org/), SciPy (http://www.scipy.org/), Pandas (http://pandas.pydata.org/) and matplotlib (http://www.matplotlib.org). All the Python scripts were developed

specifically for this study. For the fits, the least squares method was used through the 'polyfit' function of NumPy library for the linear regression and the 'curve_fit' function of SciPy library for the non-linear fits.

## 2.5 Filtering of the flux data

After the fluxes were calculated, several filters were applied to remove cases where the measurement system did not work adequately. The most common reason for discarding data was due to the problems with the chamber operation, for example, for the improper functioning of a linear actuator, which caused the chambers to remain stuck either open or closed. These cases were detected by monitoring the simultaneously measured $CO_2$ concentration data during the closure. The goodness of fit was checked by calculating the normalized root mean square error (NRMSE) (e.g. Christiansen et al., 2011; Pihlatie et al., 2013) for each fit:

$$NRMSE = \frac{\sqrt{\frac{1}{n}\sum_{i=1}^{n}(C_{fit,i}-C_i)^2}}{C_{max}-C_{min}},$$ (4)

where $n$ is the number of measurement points, $C_{fit,i}$ is the $CO_2$ concentration calculated from the fit, $C_i$ is the measured $CO_2$ concentration and $C_{max}$ and $C_{min}$ are the highest and lowest concentrations measured during closure. If the NRMSE was larger than 0.05, the $CH_4$ data from that closure were discarded. It should be noted that the application of this criterion removes closures with no change in $CO_2$ concentration, which may result when photosynthesis rate equals respiration rate. Here we found < 20 of such cases meaning that this criterion could be applied without removing a significant amount of potentially suitable data.

In addition to NRMSE filtering, the running mean of $CH_4$ flux ($F_{CH4}$) with a time window of 14 days (shifting one day at a time) and the corresponding standard deviation (σ) were calculated to remove random spiking in the data. The data points that failed to fall within $F_{CH4}\pm10\sigma$ were removed iteratively. In total, 71229 closures were recorded from which 14 % (n=9987) were discarded due to large NRMSE values (problems with the chambers) and <0.001 % (n=40) were removed with the σ-filter as outliers.

## 2.6 Detection limit

The Minimum Detectable Flux (MDF) was estimated by using the metric originally developed by Christiansen et al. (2015), which was modified by Nickerson (2016) to make it more suitable for high-frequency measurements:

$$MDF = \left(\frac{P_I}{t_c\sqrt{\frac{t_c}{p_s}}}\right)\left(\frac{VP}{ART}\right)M$$ (5)

where $P_I$ is the analytical precision of the instrument (ppm), $t_c$ is the closure time of the chamber (h), and $p_s$ is the sampling periodicity (h). The $P_I$ for the Picarro G1130 analyzer, tested and reported by the manufacturer for this specific instrument used in this study was 0.256 ppb and $p_s$ was 5 s. On a typical summer day (T=20 °C), the MDF of the system was about 0.06

µg $CH_4$ m$^{-2}$ h$^{-1}$. However, during winter the MDF was higher due to lower temperatures and the use of the extension collars which together about double the headspace volume (without snow) and therefore also the MDF.

## 2.7 The annual balance and its uncertainty

The annual balance of $CH_4$ was estimated for each chamber by first calculating the daily flux sums from the hourly fluxes and
then summing these over a year. The gaps in the data were filled by using linear interpolation between the existing hourly and daily fluxes. As most of the fluxes in 2011 and in the first quarter of 2012 were measured with a 2 min closure time, which was considered too short for the exponential regression (Sect. 3.2), we corrected the fluxes calculated with linear regression from the 2 min closures to correspond to those measured using a 6 min closure available four times a day during this period as a reference. This correction was implemented by calculating the daily median ratios between the fluxes from 6 min and 2 min
closure times, which were smoothed by a running median with a moving window of 14 days. Finally, the 2 min data from 2011 to March 2012 were multiplied by this ratio (Fig. S2).

The uncertainty of the $CH_4$ balance estimate derived from the measurements was evaluated by identifying three key error sources: (1) the random error of regression, (2) the error caused by gap filling and (3) the error caused by the correction of the fluxes measured using the 2 min closure time during the first measurement year. First, because the annual balance of each
chamber was calculated from the mean daily fluxes, we estimated the daily random error as the squared sum of the uncertainties of the hourly flux data of each day. Assuming that the goodness of fit reflects all the uncertainties related to a single flux measurement, the standard deviation of the slope estimate obtained ($\left(\frac{dC(t)}{dt}\right)_{t=0}$, Eq. 3) provides a measure of this uncertainty.

Next, the error caused by the gap filling procedure was estimated by removing one month of flux data from different parts of the whole data set and inspecting how this affected the annual balance of the different chambers. The average value of the
effect of these monthly gaps was calculated and downscaled to represent the effect of one missing day. Multiplying this value by the number of missing days during the year gives an estimate of the gap-filling error. It must be noted that the length of the removed period was similar to the longest gap observed in our data.

Last, the error estimate related to the ratio used to transform the fluxes calculated from 2 min closure to represent the 6 min closure was estimated from the median absolute deviation assuming normally distributed medians. Finally, these three error
estimates were added together by using the standard accumulation principle of independent errors.

As our measurements started in April 2011 and ended in March 2013, exactly after two years, from now on in this paper the expressions 'first year' and 'second year' denote the time periods of 4/2011–3/2012 and 4/2012–3/2013, respectively.

# 3 Results

## 3.1 Flux calculation method

Examples of typical concentration development inside a chamber during one measurement are shown in Fig. 2, for both 'high flux' case in summer (Fig. 2a) and a 'low flux' case in winter (Fig. 2b). In summer and autumn, when the fluxes were the highest, the concentration development inside a chamber usually was not adequately approximated by a linear function and thus the slope calculated with the linear regression (Eq. 1) did not properly represent the initial 'undisturbed' slope from which the flux should be calculated. As a result, linear regression resulted in lower flux estimates for these cases than exponential regression (Eq. 2). However, during the periods when the flux approached the detection limit and the concentration data became noisier, the use of exponential regression resulted in noisier flux data. Often, exponential regression created a sharp slope at the beginning of the fit in the concentration time series that resulted in unphysically high fluxes. To be able to reliably estimate the $CH_4$ exchange for the whole range of fluxes, we determined the flux limit below which the exponential regression resulted in unreliable flux estimates and the linear fit should be preferred. This limit was estimated by comparing bin (n=500) averages of linear and exponential fluxes for the whole data set (using a 6 min closure time) (Fig. 3). When the linearly calculated fluxes fell below ca. 2.5 µg $CH_4$ m$^{-2}$ h$^{-1}$, the noise in the flux calculated using the exponential regression increased steeply and the shape of the relationship changed (Fig. 3). Therefore, we decided to first calculate the flux with the linear regression and to recalculate all the fluxes exceeding the limit of 2.5 µg $CH_4$ m$^{-2}$ h$^{-1}$ with the exponential regression. Henceforth, all the data shown in this paper have been calculated in this way unless stated otherwise.

The whole 2-year data set showed that the $CH_4$ fluxes calculated with linear regression (Eq. 1) were systematically and significantly lower than those calculated with exponential regression (Eq. 2) (Table 3). The seasonal average flux difference between the linear and exponential regressions varied within 10.9–44.4 % (average over 2 yr 27.5±0.3 %, ±95 % confidence interval). The mean relative difference was dependent on the time of the year: it was largest during the winter and spring (24.9–44.4 %) when the soil $CH_4$ sink was at its lowest, and smallest in summer and autumn (10.9–14.4 %) when the sink was at its highest. When comparing individual measurements, the average relative difference between the linear and exponential regression was slightly smaller in 2012 compared to 2011. Also, the uncertainties associated with the fluxes were slightly larger in 2011 probably due to the fewer measurements available with 6 min closure time.

## 3.2 Effect of closure time on fluxes

The effect of the different fitting time windows was tested by both increasing the fitting period from the beginning of the closure with 10 s steps and by keeping the fitting window constant but moving its starting point. For these tests, we used the data from summer 2012, when the measurements were made with a 16 min closure time. The flux from the exponential fit was clearly affected by the length of the fit when the fitting period was < 190 s (Fig. 4b). After that, the mean difference was mostly statistically insignificant (p > 0.05), as compared to the flux calculated with the 900 s period. On the other hand, the estimated linear flux stayed about the same for the first 140 s resulting in 16.2±0.6 % higher fluxes than obtained with the 900 s fitting

window (Fig. 4a). However, further increase of the fitting period systematically decreased the estimated flux by about 1.3 % per 60 s. A decrease of $17.3 \pm 3.0$ % was also observed when the starting point of the fit was delayed by 530 s, but the fitting period was kept constant at 6 min (Fig. 5).

Even though the results above might support the selection of a fitting period of 190 s, a 6 min fitting period was applied in further analysis. This was selected based on three arguments: (1) it made the exponential regression results more stable; (2) we wanted to use the same fitting period in both linear and exponential regressions, and (3) a longer fitting period decreases the detection limit (Eq. 5). The last point was mainly related to winter measurements when the detection limit was increased by lower temperatures and the use of extension collars (increasing the effective volume before the collars were filled with snow). However, in 2011 and in the first quarter of 2012, a 2 min closure time was mostly used, which proved to be too short for accurate estimates with the exponential fit. As a result, the results from these shorter measurements were corrected to correspond to those obtained with the 6 min closure time (Sect. 2.7).

### 3.3 Seasonal dynamics of $CH_4$ flux and comparison between the chambers

During the 2-year measurement period, $CH_4$ fluxes varied mainly between $-120$ and $+20$ µg $CH_4$ m$^{-2}$ h$^{-1}$ (Figs. 6 and 7). Higher uptake rates (70 µg $CH_4$ m$^{-2}$ h$^{-1}$) were measured more often during the first year (Fig. 7a) and during summer 2011 (Fig. 7b) than in the second year and in summer 2012. The number of emission cases was low in both years and the soil acted as a $CH_4$ sink for most of the time in all chambers, although a few emission peaks of up to 200 µg $CH_4$ m$^{-2}$ h$^{-1}$ were recorded during and following heavy rainfall events. The data filtering (Sect. 2.5) was designed to be flexible to prevent the removal of these short-lasting $CH_4$ bursts from the accepted data set. While some emission peaks were observed in most of the chambers, the peaks were largest in chamber #6. However, the peaks did not necessarily occur at the same time in different chambers.

$CH_4$ fluxes showed a clear seasonal variation. In spring, when the snow melted and thawing of the soil surface had started, $CH_4$ emissions fluctuated around 2 µg $CH_4$ m$^{-2}$ h$^{-1}$ in all chambers (Fig. 6). As the temperature rise continued, increasing $CH_4$ uptake was observed. In both years, $CH_4$ uptake was largest in August, when fluxes varied between $-30$ and $-130$ µg $CH_4$ m$^{-2}$ h$^{-1}$. In September, the uptake decreased and by the end of November it had dropped to half of that in the summer. However, the soil acted as a sink until the soil surface froze, after which $CH_4$ fluxes fluctuated around zero.

Considerable systematic differences in fluxes between the chambers were detected. The largest two-year average sinks were measured in chambers #5 and #2 (Fig. 8), which were dominated by forest mosses *Pleurozium schreberi* and *Dicranum polysetum* (Table 2). The third largest sink was observed in chamber #6 (Fig. 8) with a *Sphagnum* sp. carpet, while the remainder of the chambers showed similar fluxes. Annual net $CH_4$ exchange rates were on average $-267 \pm 55$ and $-172 \pm 31$ mg $CH_4$ m$^{-2}$ yr$^{-1}$ ($\pm$Standard error of the mean) in 2011–2012 and 2012–2013, respectively. The forest floor sink was 10–44 % lower during the second than first annual period and this difference was statistically significant ($p < 0.05$) for all the chambers except chamber #4. The largest absolute year-to-year reduction in annual $CH_4$ exchange was observed in the two chambers dominated by *Pleurozium schreberi* and *Dicranum polysetum* (#5 and #2), followed by the chamber dominated by *Sphagnum*

sp. (#6). However, the largest relative decrease in net $CH_4$ exchange happened in chamber #6 (44 %) while in the rest of the chambers the decrease varied within 10–37 %.

The sink period, i.e. the period when all chambers acted as $CH_4$ sinks (daily mean flux $< -2$ µg $CH_4$ m$^{-2}$ h$^{-1}$), was slightly longer in the first than in the second year. In the first year, all chambers acted as sinks from 14 May 2011 to 20 February 2012, a total of 282 days. During the second year, the sink period lasted for 269 days from 10 May 2012 until 3 February 2013. In addition, chamber #6 acted as a sink over the whole winter and spring in 2012, while from chambers #2 and #5 daily mean fluxes of $< -2$ µg $CH_4$ m$^{-2}$ h$^{-1}$ were measured already at the start of the study in April 2011.

### 3.4 Factors controlling the short- and long-term variations in $CH_4$ exchange

### 3.4.1 Seasonal cycle

There was an observable, exponential relationship between the mean daily $CH_4$ fluxes and the deeper soil temperatures when the data from the spring and summer of 2011 were pooled (Fig. 9a). Splitting these data into shorter periods showed that the relationship was rather strong in April–May and in June–early July, but after that, when the soil temperature at 30 cm depth exceeded 12°C, the relationship was absent. It is evident that there is some covariation between soil temperature and WTL with typically higher $CH_4$ uptake taking place at lower WTL. The plot between the residuals of the temperature response against WTL (Fig. 9b) suggests that the variation in the $CH_4$ uptake during the latter half of July and August was better explained by the WTL than temperature. All the chambers recorded similar behaviour in 2011 and 2012, although in 2012 the data were noisier and the relationships observed in 2011 were not as clear.

### 3.4.2 Diurnal cycle

In addition to seasonal dynamics, diurnal variation was observed in $CH_4$ fluxes at least occasionally in all chambers, mostly in May and in the first half of June. Such variation was more common in 2012 and was observed in all of the chambers, while in 2011 the variation occurred mostly in chambers #2, #5 and #6. For example, during the first two weeks of June 2011 a clear diurnal cycle coinciding with the variation in air temperature was observed (Figs. 10 and 11a–b). Higher $CH_4$ uptake was observed during the night and morning hours, while the sink decreased towards the midday and started to grow again towards the night (Fig. 11a). This particular 2-week period was associated with high daytime air temperatures, reaching almost +30°C, and a relatively low WTL (Figs. 10 and 11a–b). After that, at about mid-June, the weather type changed to cool and wet and the diurnal variation was diminished or absent (Figs. 10 and 11d–e). Such behaviour, illustrated in Fig. 10, was typical for the rest of the growing season data: diurnal variation occurred more often with dry weather, while during and after the rain the variation ceased for a while. However, WTL itself did not have an impact on the diurnal cycle.

Pooling the data of these 2-week periods into hourly means implies that the soil temperature may exert a strong control on the $CH_4$ exchange (Fig. 11). However, as temperature often – though not always – tends to correlate with wind speed, particularly in the summer, it is necessary to consider the effect of both these variables. The correlations calculated from hourly data

indicate that it was the wind speed and not the temperature which played the major role in causing the diurnal variation in $CH_4$ flux (Figs. S6-S27). However, this relationship was not comprehensive, as the correlation with wind speed was absent during some periods. In qualitative terms, we observed that the drier the soil was, the greater was the impact of wind speed.

To explain this correlation, we investigated the relationship between the parameter $c_{exp}$ (Eq. 2) and ambient wind speed, in 2-week to one month periods to diminish the possible impact of seasonality (Figs. S28-S40). $c_{exp}$ represents the curvature of concentration time series during each chamber closure. For example, the $c_{exp}$ determined for chamber #2 became less negative when wind speed exceeded 2 m s$^{-1}$. This means that the curvature was weaker for cases of high wind speed.

### 3.4.3 Correlation between the flux and meteorological variables

To understand the general driving factors behind all the variation, either diurnal or seasonal, Pearson correlation coefficients between the hourly $CH_4$ flux and key environmental variables were calculated for different seasons, averaging the correlations determined separately for each chamber (Table 4).

There was a highly significant negative correlation between $CH_4$ flux and soil temperature (i.e. higher uptake at higher temperature). The deeper soil temperatures (at 20 cm and 30 cm depths) showed the best correlation in all seasons. Also, T5 and T2 often correlated with $CH_4$ flux. The highest correlations were observed in springs, winters and in autumn 2012, and all these correlations were significant in all the chambers with the exception of summer 2011. Also, the correlations of ambient (AirT) and soil surface temperature (ST) were systematically lower than those of the deeper soil temperatures. The correlation with surface soil temperatures was absent or very low in summer 2011, when only the deeper soil temperatures showed a significant correlation.

WTL and $CH_4$ flux correlated positively in both summers (i.e. the deeper the WTL, the higher the uptake), while a negative correlation, (i.e. the deeper the WTL, the higher the emission) was found for the winters. The correlation between the flux and PAR was always low ($|r|<0.2$) or absent.

A significant positive correlation between wind speed (WS) and the flux was found in both summers and autumns and in spring 2012, meaning that the estimated sink decreased when wind speed increased. This correlation was especially clear in the time periods when there was diurnal variation in the flux, but it was non-existent when no diurnal cycle was observed, which is consistent with the result reported above. However, even though the correlation was high in autumn 2011, no statistically significant diurnal variation in the flux was observed at that time (data not shown). Of all the inspected quantities, wind speed was the best explanatory factor of the diurnal cycle of $CH_4$ flux; this was followed by temperature quantities.

## 4 Discussion

### 4.1 Chamber closure time and flux calculation method

In this study, we found that using the linear regression method in flux calculations resulted in 20–50 % lower flux estimates for most of the time, in comparison to fluxes calculated with an exponential fit. On the other hand, in winter and early spring,

i.e., at the time of low absolute $CH_4$ fluxes (< 2.5 µg $CH_4$ $m^{-2}$ $h^{-1}$), linear regression gave more reliable results with clearly lower hour-to-hour noise in the fluxes. The uncertainty associated with exponential regression with low fluxes was caused by the decreased signal-to-noise ratio in the concentration data, leading to more or less arbitrary values of the concentration change estimated at t=0. This explains why the difference between the fluxes estimated by the linear and exponential models was highest during winter and spring.

The use of exponential regression can be considered especially justified during summer and autumn when the concentration development inside the chamber is strongly non-linear. During these periods, linear regression gave significantly lower flux estimates with an average difference of 12.8±1.8 %. In winter and spring, however, the average seasonal difference was as high as 44 %. Concerning the annual balance, it should be noted that this difference has a greater influence during summer and autumn when the fluxes are up to two orders of magnitude larger. The mean difference during the whole measurement period (27.5±0.3 %) is in agreement with many previous studies; for example, Anthony et al. (1995) and Pedersen et al. (2010) reported a 35 % and 34 % decrease in flux values, respectively, when using linear regression instead of exponential regression. Kutzbach et al. (2007) reported that underestimations for multiple sites varied mostly within 20–60 %, while Pihlatie et al. (2013) found an average underestimation of 30 % under laboratory conditions at different flux levels. However, the above-mentioned studies focused on either $CH_4$ and $CO_2$ emissions or $CO_2$ uptake and did not include measurements related to a small soil sink of $CH_4$. Since the exponential fit results in larger uptake estimates, which we consider to better represent the correct flux at the time of the chamber closure, we agree with the previous studies that recommend the use of exponential regression in flux calculation. However, due to the considerable noise generated for small fluxes, we recommend that in the future studies employing temporally high-resolution data, the fluxes should be calculated initially with both methods to determine the threshold for "low" fluxes. As this threshold value is dependent on the measurement system and method, it cannot be generalized.

We found that increasing the fitting period up to 15 min from the beginning of the concentration time series systematically decreased the flux estimated by the linear regression model, but only after this period exceeded 140 s. On the other hand, the flux estimated by the exponential regression decreased until the fitting window covered 190 s after which the estimated flux was stabilized. To disentangle whether these anomalous patterns in the beginning of both curves were caused by an initial disturbance to the measurement or if they represented a real phenomenon affecting the flux, we removed 2 min of data from the beginning of concentration time series. This removed the plateau with large variation from the linear fit results, but did not change the shape shown for the exponential fit in Fig. 4b (results not shown). This suggests that the concentration data during the first minute of measurement are perturbed by chamber closure and that the linear fit is more sensitive to this than the exponential fit, when using a short fitting period. On the other hand, the exponential fit seems to overestimate the flux if the fit is limited to a short time window, but this overestimation is not related to a possible disturbance in the beginning of the data.

Because the concentration development is non-linear, it is not only the length of the fitting period, but also the start of this period which is important for the flux estimate. The more data are removed from the beginning, the smaller the estimated $CH_4$

uptake becomes. Ueyama et al. (2015) noticed that the flux estimates by linear and exponential regressions both decreased with increasing closure time. Their slopes of linear regression ($14\pm2$ % over 5 min) decreased faster than those in our study ($16.2\pm0.2$ % over 13 min), suggesting that their concentration data were more non-linear. The difference could be partly explained by the fact that Ueyama et al. (2015) used smaller chambers and measured higher fluxes which both result in a faster
decrease of the vertical concentration gradient between the soil and the chamber headspace.

## 4.2 CH$_4$ exchange dynamics in a peatland forest

### 4.2.1 Annual balances

The measurement site (excluding the ditches) was a small annual CH$_4$ sink (varying from $172\pm31$ to $267\pm55$ mg CH$_4$ m$^{-2}$ yr$^{-1}$, $\pm$Standard error of the mean) over the two-year measurement period. While we do not have measurements prior to drainage,
it is possible to roughly estimate the pre-drainage fluxes from measurements conducted at similar sites. As Lettosuo was originally a herb-rich tall sedge birch-pine fen, it can be considered similar to the site reported by Nykänen et al. (1998). That site has high CH$_4$ emissions of 25 g CH$_4$ m$^{-2}$ yr$^{-1}$, so it is obvious that the drainage of Lettosuo has turned the peat soil from a CH$_4$ source to a small CH$_4$ sink. However, it should be noted that our calculations of annual and daily CH$_4$ exchange do not include the emissions from the ditches, which have been found to be highly variable: from 0 to 600 mg CH$_4$ m$^{-2}$ day$^{-1}$ (e.g.
Minkkinen et al., 1997; Minkkinen and Laine, 2006; Luan and Wu, 2015). At Lettosuo, where the ditches cover 2–3 % of the area, we estimated, based on 41 manual chamber flux measurements from six points made in the latter half of 2011, that CH$_4$ emissions (per m$^2$ ditch) from the ditches averaged 22 ($\pm$38, $\pm$SD) g CH$_4$ m$^{-2}$ yr$^{-1}$ (unpublished data). Simple upscaling suggests that, when ditches are accounted for, Lettosuo is a small annual source of CH$_4$ to the atmosphere, although the high uncertainties associated with this calculation should be noted.
Previous studies of drained peatland forests in Finland have reported uptake rates varying between 10 and 970 mg CH$_4$ m$^{-2}$ yr$^{-1}$ (Alm et al., 1999; Minkkinen et al., 2007; Ojanen et al., 2010; Lohila et al., 2011). The average annual uptake in boreal upland forests typically varies from about 100 to 500 mg CH$_4$ m$^{-2}$ yr$^{-1}$ (Smith et al, 2000; Dutaur and Verchot, 2007; Lohila et al., 2016), but also annual net emissions from upland forests have been reported (e.g. Sundqvist et al., 2015; Lohila et al., 2016). Thus, the net annual CH$_4$ exchange measured at Lettosuo (excluding the ditches) was well within the typical range of
the average CH$_4$ sinks reported for boreal upland forests.

### 4.2.2 Emission peaks

A few larger CH$_4$ emission peaks were observed during and after heavy rainfall events in summer, but not all chambers responded to the same rainfall events. The largest number and magnitude of these short bursts of CH$_4$ were recorded with the *Sphagnum*-dominated chamber (#6), which was expected as *Sphagnum* mosses favor wet spots. These rainfall events turned
the soil from a sink to a source (up to 200 µg CH$_4$ m$^{-2}$ h$^{-1}$ in chamber #6) for a short period ranging from a few hours to few

days. This was possibly due to increased water saturation and decreased air-filled pore space leading to reduction in oxygen diffusion, which could promote methanogenic activity and suppress methane oxidation.

Similarly to our results, Nykänen et al. (1995) observed that a drained peatland soil can switch to a $CH_4$ source (up to 200 µg $CH_4$ $m^{-2}$ $h^{-1}$) during water saturation event. However, the emission peaks at Lettosuo were relatively small when compared to

upland mineral soil forest sites, where emissions of up to 3.7 mg $CH_4$ $m^{-2}$ $h^{-1}$ have been observed in wet conditions (e.g. Savage et al., 1997; Lohila et al., 2016). Similarly, the maximum hourly uptake in summer at Lettosuo was rather similar to the fluxes reported for the above-mentioned upland forests (from –50 to –120 µg $CH_4$ $m^{-2}$ $h^{-1}$ at Lettosuo vs. –40 to –80 µg $CH_4$ $m^{-2}$ $h^{-1}$ at upland forests).

### 4.2.3 Spatial variation

There were relatively large differences in the annual net $CH_4$ exchange rates between the chambers (from –104 mg $CH_4$ $m^{-2}$ $yr^{-1}$ in chamber #3, to –505 mg $CH_4$ $m^{-2}$ $yr^{-1}$ in chamber #5), even though all the chambers were located within a maximum distance of about 15 m from each other. The difference in the soil surface temperature between the chambers was usually less than 2 K, which indicates that the soil temperature was not the main factor determining the observed spatial variation in fluxes. Since we do not have WTL data for each of the chambers separately, we cannot quantitatively evaluate its role in the spatial

variation of fluxes. It is unlikely that the variation in the WTL solely could explain the difference: even though chambers #4 and #5 were located at the same distance from a ditch, and probably had the deepest WTL, chamber #5 showed the highest uptake, while chamber #4 was one of the smallest sinks. Hence, although WTL is likely to explain part of the spatial variation in the $CH_4$ fluxes, there are potentially many other factors, such as the vegetation composition and small-scale soil properties. The smallest sink was observed in chambers #1, #3 and #4, which were characterized with the lowest (#3) and highest (#1, #4)

vascular green area ($VGA_{max}$) values. Thus it seems that it was not the amount of ground vegetation which affected the sink, but a more relevant factor could have been the coverage of mosses vs. that of vascular plants within the collar, especially that of the forest mosses *Pleurozium schreberi* and *Dicranum polysetum*, which were particularly abundant in the highly oxidizing chambers #2 and #5. Due to the small number of chambers, however, the relationship between the forest floor vegetation and the $CH_4$ exchange may be coincidental and can only be speculated.

### 25  4.2.4 Diurnal variation

All the chambers recorded diurnal variation in $CH_4$ flux at some time during the study period with most of the variation observed during late spring and early summer. Typically, $CH_4$ uptake was at its highest during the night, and decreased towards the afternoon. The diurnal variation was more common and occurred more often in all chambers in 2012, while in 2011 it occurred mostly in chamber #2. This variation usually ceased or was at least greatly diminished during and after rainfall events,

but usually it appeared again after a couple of days. WTL as such, however, did not have an impact on the diurnal cycle, which suggests that the conditions in the soil surface were much more important for this phenomenon.

Although the diurnal variation seemingly followed the patterns in the air and soil surface temperatures, it was best explained by the wind speed (WS) measured above the canopy (below canopy WS is not available). To study further this relationship, we tested the correlation between the parameter $c_{exp}$ (Eq. 2) and WS. $c_{exp}$ represents the curvature in the exponential fit, being negative whenever the concentration increase during a chamber closure shows a slowing shape. As we only selected negative,

i.e. uptake, fluxes here, it follows that a more negative $c_{exp}$ indicates a higher curvature in the concentration evolution. Should leaking be responsible for the smaller $CH_4$ uptake during daytime, as the observed relationship between WS and $CH_4$ exchange implied, it would be logical to find higher curvatures with higher WS. Such a relationship was recently found by Pirk et al. (2016) for $CH_4$ emission chamber flux data from pristine peatlands. However, we did not observe such a relationship in our $CH_4$ uptake data. For example, in chamber #2, in which the diurnal cycle was most explicit, an increasing $c_{exp}$ was determined

for most of the periods studied. There were only a few chambers and periods when $c_{exp}$ decreased with increasing WS. Thus we must conclude that the diurnal variation in our data is related to the technical operation of the chamber rather than environmental conditions. Nevertheless, as the temperature and WS correlated strongly, it is possible that some of the observed pattern was due to some microbial or environmental factor.

We hypothesize that, rather than chamber leaking, the main underlying factor for the clear negative correlation between wind

speed and $CH_4$ uptake is related to changes in the soil storage and thus the changes imposed by chamber closure to the concentration gradient within the top soil and the adjacent air layer. Prior to the closure, this gradient is controlled by atmospheric mixing and hence strongly affected by the ambient wind speed. During a calm night with a cool soil surface, turbulent mixing is strongly suppressed and molecular diffusion gains importance, while windy and sunny conditions result in much smaller vertical gradients due to vigorous turbulence that is also able to perturb the top-soil pore space. After the chamber

is closed, the concentration gradient adjusts to the constant mixing generated by a fan. Thus, the change in concentration gradient depends on the mixing conditions that prevail above the target surface just before the chamber is introduced and how these relate to the mixing rate of the chamber headspace air. In the nocturnal case outlined above, mixing is enhanced after the chamber closure, resulting in a higher $CH_4$ uptake in the chamber.

The absence of the diurnal cycle in winter, and during and after the rain in summer, can be explained by the increased soil

moisture content, which decreases the air-filled pore space in soil, thus hampering the wind-induced mixing effect at the soil–atmosphere interface and by slowing down the diffusion rate (Pirk et al., 2016).

It should be noted that the situation is different when $CH_4$ exchange is measured above a forest canopy with the EC method. In that case, the measurement does not significantly disturb atmospheric mixing and increased mechanical turbulence potentially enhances vertical gas exchange. Such positive correlation between the downward $CH_4$ flux and wind speed, with

higher sinks during the daytime, has been reported by Wang et al. (2013). This is consistent with the results of our fan-speed test, described in Koskinen et al. (2014) who measured $CO_2$ respiration by the same chamber system. The $CH_4$ flux data from the same test showed a higher $CH_4$ uptake with higher fan speed (data not shown).

A wind-induced diurnal cycle suggests that the current chamber set-up potentially leads to an over- or underestimate of the actual uptake rate during lower or higher fan-induced mixing, respectively, as compared to ambient mixing by wind. The

chamber construction could be improved by making the fan speed vary as a function of the ambient wind speed, so as to mimic the variations in atmospheric mixing. However, we can expect that the systematic bias resulting from the wind response is minimized when employing automated sampling that facilitates continuous measurements. Our results imply that sporadic sampling with manual chambers, which is typically limited to the daytime, would have resulted in lower uptake estimates for

this site than the extensive data collected with our automatic system.

## 4.3 Driving factors for the CH$_4$ efflux

The seasonal CH$_4$ fluxes correlated best with soil temperatures at the depths of 20 cm and 30 cm, but significant correlations occurred also with soil temperatures at other depths for most of the study period. The correlations with air and soil surface temperatures were lower. The correlations were always negative, indicating that higher temperatures promoted the soil CH$_4$

uptake. This observation could be attributed to increased consumption of CH$_4$ by methanotrophs in higher temperatures that enhance methanotrophic activity (e.g. Mohanty et al., 2007). However, it is likely that in addition to – or even instead of – the increased methanotrophic activity, there are other reasons behind this relationship. The covariation of temperature with other variables, such as ground water level and phenology, all typically peaking in July–August, may lead to spurious correlation between temperature and CH$_4$ flux. Indeed, the flux was also correlated with WTL, the correlation being significantly positive

(higher uptake with lower WTL) in spring, summer and autumn, but negative in winter. At our site, the soil layers most favourable for methane production and oxidation are located at clearly different depths in the soil, the first being found below the water table and the latter much closer to the soil surface (A. Putkinen, unpublished data). Both of these have distinct temperature and moisture responses, which are practically impossible to disentangle by examining the net CH$_4$ flux observed at the surface.

In addition to the correlations found in the hourly data, we found evidence that lowering WTL increases the daily CH$_4$ uptake in the latter part of summer, when WTL < –40 cm. In the beginning of the summer, the daily fluxes were better explained by the soil temperature, while after the mid-July the WTL overshadowed the temperature as a control of the daily fluxes.

In pristine peatlands, temperature has been shown to correlate positively with the CH$_4$ emission rate (e.g. Mikkelä et al., 1995; Bellisario et al., 1999; Mastepanov et al., 2013). In drained peatland forests, significant correlations between CH$_4$ flux and

temperature have been found in CH$_4$-emitting ecosystems (Nykänen et al., 1998; Minkkinen and Laine, 2006), although the direction of the correlation has been found to differ between fens and bogs. In contrast, no significant correlations with temperature were found in peatland forests that mainly showed CH$_4$ uptake (Ojanen et al., 2010; Wang et al., 2013). Both CH$_4$ emission and uptake have been found to correlate with WTL (e.g. Bellisario et al., 1999; Ojanen et al., 2010).

In addition to temperature, WTL and WS, CH$_4$ flux has been found to correlate with tree stand volume (Ojanen et al., 2010;

Minkkinen et al., 2007), which is, on the other hand, an indirect measure of the WTL. Also, PAR has been observed to correlate positively with CH$_4$ emissions in a *Sphagnum*-dominated mixed mire (Mikkelä et al., 1995). In this study, however, the correlations with PAR were low or absent.

**5 Conclusions**

In this paper, we have presented a two-year data set of $CH_4$ exchange measured at the forest floor of a boreal forestry-drained fen. These results show that automated chamber measurements with an accurate on-line gas analyzer make it possible to observe small $CH_4$ fluxes even during the winter with snow cover. Although the fluxes at our site were relatively low

throughout the year, we succeeded in catching the annual cycle in the $CH_4$ uptake. Our results indicate that the forest floor of this peatland site acted as a small annual $CH_4$ sink (mean balance $-219$ mg $CH_4$ m$^{-2}$ yr$^{-1}$), although completing the balance with the emissions from ditches indicates that the site is likely to be a small $CH_4$ source.

In spite of the low flux detection limit of the measurement system, our results indicate that it is necessary to pay attention to the flux calculation methods, and instead of choosing between linear and exponential fits we decided on a combination of both.

Even though the fit based on linear regression was observed to typically result in a smaller flux than an exponential fit, its use was justified for low fluxes by showing that it produced more robust estimates when the concentration change during chamber closure was small and thus more affected by measurement noise. In addition, we demonstrated that both the length of the fitting period and the starting time of this window had a significant effect on the flux estimates and thus cannot be selected arbitrarily. The $CH_4$ uptake, measured with closed chambers, was observed to correlate with wind speed, which caused a corresponding

diurnal cycle. However, this was partly attributed to aerodynamic effects due to chamber closure, which are dependent on atmospheric mixing prior to the closure. Thus, the chamber construction could be potentially improved by adjusting the chamber fan speed according to the ambient wind speed. As this variation is partly related to changes in the soil $CH_4$ storage, the error introduced in the annual balance estimated from short-term fluxes can be diminished by continuous measurements fully covering the diurnal cycle. Continuous long-term measurements also facilitate the analysis of the environmental factors

that control $CH_4$ exchange. However, in order to understand the biological processes involved in $CH_4$ production and oxidation, i.e. the processes behind the net $CH_4$ flux observed, additional measurements are necessary, focusing on the production and oxidation potentials and the within-soil gas gradients.

Since the considerations of the measurement system performance are site- and system-specific, we recommend that any future study should address the procedures involved in flux calculation, including the fitting method and the length and delay of the

fitting period, based on the analysis presented above. In particular, we recommend using the flux limit method applied in this study, i.e. using linear regression for low fluxes and exponential regression for fluxes above a threshold to be determined.

**Data availability**

The measured flux and meteorological data will be made available through European Fluxes Database Cluster (http://gaia.agraria.unitus.it/home).

**Acknowledgements**

We are grateful for the financial support from the Maj and Tor Nessling foundation and from the Ministry of Transport and Communications through the Integrated Carbon Observing System (ICOS) research. We would also like to thank Pentti Arffman and Tero Hirvonen for their help in data treatment and for measurements at the site.

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

Table 1. CN ratio, bulk density and ash content (±SD) of the peat at Lettosuo (n=4).

|  | CN ratio | Bulk density (g cm$^{-3}$) | Ash content (%) |
|---|---|---|---|
| **Humus** | 29.2 ± 1.8 | 0.005 ± 0.003 | 3.1 ± 0.4 |
| **0–10 cm** | 23.9 ± 1.0 | 0.107 ± 0.014 | 6.5 ± 1.5 |
| **10–20 cm** | 24.3 ± 0.7 | 0.170 ± 0.011 | 3.4 ± 0.4 |

Table 2: Ground vegetation, all-sided maximum vascular green area (VGA$_{max}$, m$^2$ vascular green surface m$^{-2}$ forest floor) and coverage (%) of forest mosses (C$_{FM}$) and *Sphagnum* mosses (C$_{SP}$).

| Chamber | Vegetation | VGA$_{max}$ | C$_{FM}$ | C$_{SP}$ |
|---|---|---|---|---|
| 1 | *Pleurozium schreberi*<br>*Dicranum polysetum*<br>*Vaccinium myrtillus* | 2.04 | 56 | 0 |
| 2 | *Pleurozium schreberi*<br>*Dicranum polysetum*<br>*Vaccinium vitis-idaea* | 0.85 | 60 | 0 |
| 3 | *Maianthemum bifolium*<br>*Pleurozium schreberi*<br>*Dicranum polysetum* | 0.01 | 3 | 0 |
| 4 | *Dryopteris carthusiana*<br>Vaccinium myrtillus<br>*Vaccinium vitis-idaea*<br>*Pleurozium schreberi*<br>*Dicranum polysetum* | 2.34 | 26 | 0 |
| 5 | *Pleurozium schreberi*<br>*Dicranum polysetum* | 0.11 | 30 | 0 |
| 6 | *Sphagnum girgensohnii* | - | 0 | 90 |

Table 3: a) Seasonal (summer=JJA, autumn=SON, winter=DJF, spring=MAM) averages of $CH_4$ flux ($\mu$g $CH_4$ m$^{-2}$ h$^{-1}$) calculated with linear ('linear flux') and exponential regression ('exponential flux') with 95 % confidence intervals ($\pm$). The difference shows how much smaller the linear flux was on average when compared to the exponential flux. The data include the fluxes measured by all six chambers with a 6 min closure time.

| | Summer 2011 | Summer 2012 | Autumn 2011 | Autumn 2012 | Winter 11-12 | Winter 12-13 | Spring 2011 | Spring 2012 | 4/11-3/13 |
|---|---|---|---|---|---|---|---|---|---|
| **Mean linear flux** | -38.9 ±1.3 | -28.7 ±0.6 | -39.0 ±1.2 | -31.3 ±0.4 | -18.1 ±0.7 | -4.5 ±0.1 | -6.8 ±0.5 | -9.5 ±0.4 | -18.6 ±0.2 |
| **Mean exponential flux** | -45.1 ±1.5 | -32.2 ±0.7 | -44.9 ±1.3 | -34.9 ±0.4 | -21.0 ±5.3 | -4.8 ±5.2 | -25.9 ±26.5 | -12.3 ±1.2 | -22.3 ±0.4 |
| **Number of closures** | 1558 | 3685 | 1477 | 10642 | 1597 | 12209 | 3180 | 10272 | 48182 |
| **Difference (%)** | 14.4 ±0.5 | 11.7 ±0.3 | 14.2 ±0.5 | 10.9 ±0.2 | 24.9 ±1.0 | 38.9 ±0.8 | 44.4 ±1.5 | 38.2 ±1.0 | 27.5 ±0.3 |

Table 4: Correlation coefficients (r) between hourly CH$_4$ flux and environmental variables averaged for six chambers during different seasons. **Bolded and underlined:** all six chambers with $p < 0.01$; **bolded:** five chambers with $p < 0.01$, underlined: four chambers with $p < 0.01$; *italics*: three chambers with $p < 0.01$. AirT=air temperature, ST=soil surface temperature, PAR=photosynthetically active radiation, WS=wind speed measured above the canopy, Tx=soil temperature at a depth of x cm, WTL=water table level. For clarity, negative r-values typically denote situations where CH$_4$ uptake increases when the value of explaining variable increases. The season definitions are the same as in Table 3.

| Season | AirT | ST | PAR | WS | T2 | T5 | T10 | T20 | T30 | WTL[*] |
|---|---|---|---|---|---|---|---|---|---|---|
| **Spring 2011** | -0.03 | **-0.04** | *-0.18* | -0.08 | **_-0.41_** | **_-0.52_** | **_-0.61_** | **_-0.67_** | **_-0.70_** | **_0.52_** |
| **Summer 2011** | -0.05 | *0.05* | **0.17** | **_0.30_** | 0.08 | **-0.26** | **-0.37** | **_-0.54_** | **_-0.61_** | **_0.60_** |
| **Autumn 2011** | **_-0.19_** | **_-0.16_** | 0.16 | **_0.40_** | **_-0.19_** | **_-0.19_** | **_-0.2_** | **_-0.21_** | **_-0.22_** | **0.39** |
| **Winter 11-12** | **_-0.34_** | **-0.23** | -0.17 | *-0.05* | **_-0.70_** | **_-0.75_** | **-0.59** | **_-0.83_** | **_-0.83_** | **_-0.83_** |
| **Spring 2012** | **_-0.48_** | **_-0.57_** | 0.00 | **_0.15_** | **_-0.69_** | **_-0.74_** | nd | **_-0.82_** | **_-0.82_** | **_0.47_** |
| **Summer 2012** | -0.16 | -0.17 | 0.14 | **_0.21_** | **_-0.30_** | **_-0.42_** | nd | **_-0.60_** | **_-0.65_** | **_0.59_** |
| **Autumn 2012** | **_-0.47_** | **_-0.53_** | 0.11 | **_0.28_** | **_-0.58_** | **_-0.61_** | **_-0.45_** | **_-0.64_** | **_-0.64_** | **_0.60_** |
| **Winter 12-13** | **_0.29_** | **_0.37_** | *0.07* | 0.02 | **_-0.37_** | **_-0.73_** | **_-0.79_** | **_-0.79_** | **_-0.79_** | **_-0.35_** |

*Negative WTL denotes water level below the soil surface, i.e. positive correlation results from increasing uptake with decreasing WTL.

nd = not determined

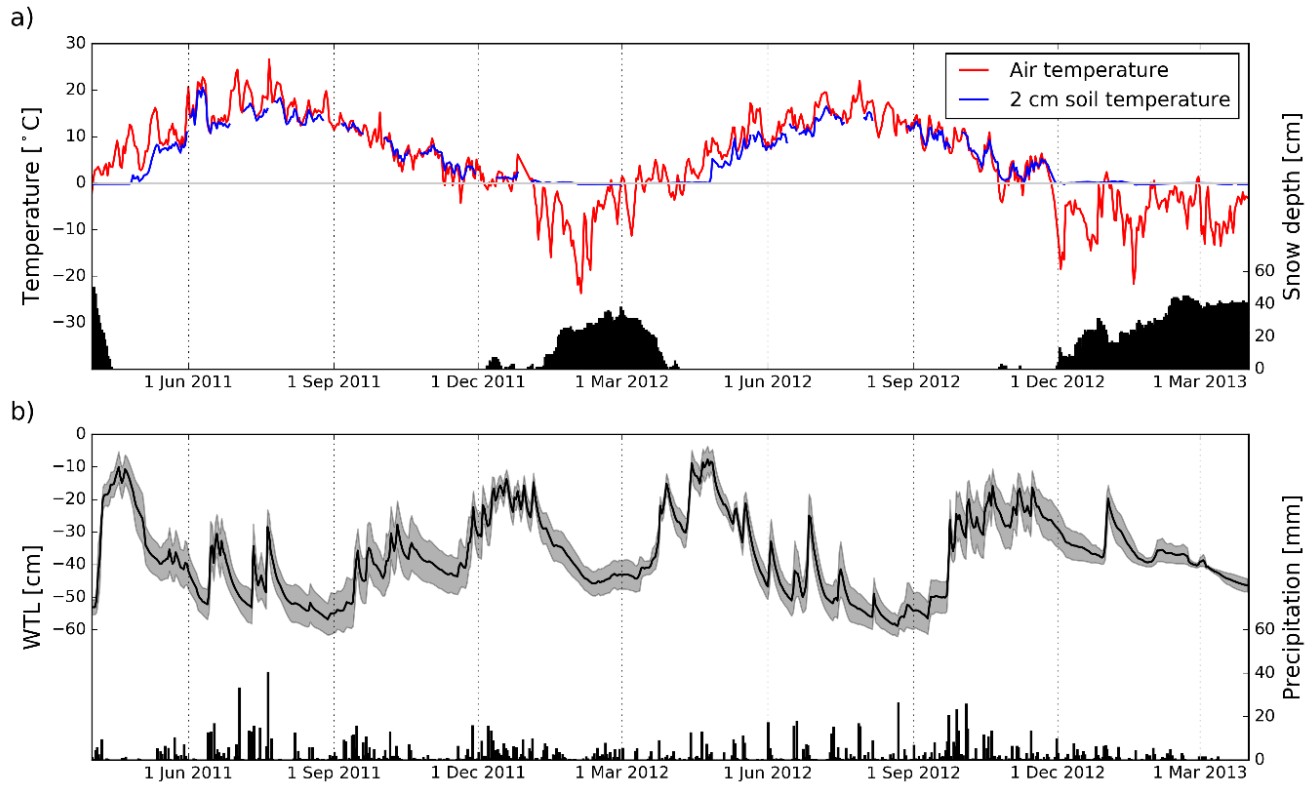

**Figure 1: (a) The daily mean of air temperature (red) and soil temperature at 2 cm depth (blue) at Lettosuo during the measurement period (1 April 2011 to 31 March 2013), and the daily snow depth (bars) measured at the nearby Jokioinen observatory. (b) The daily mean water table (WTL) (line) and its standard deviation (shading) from four different points at Lettosuo and the daily precipitation (bars) measured at Jokioinen.**

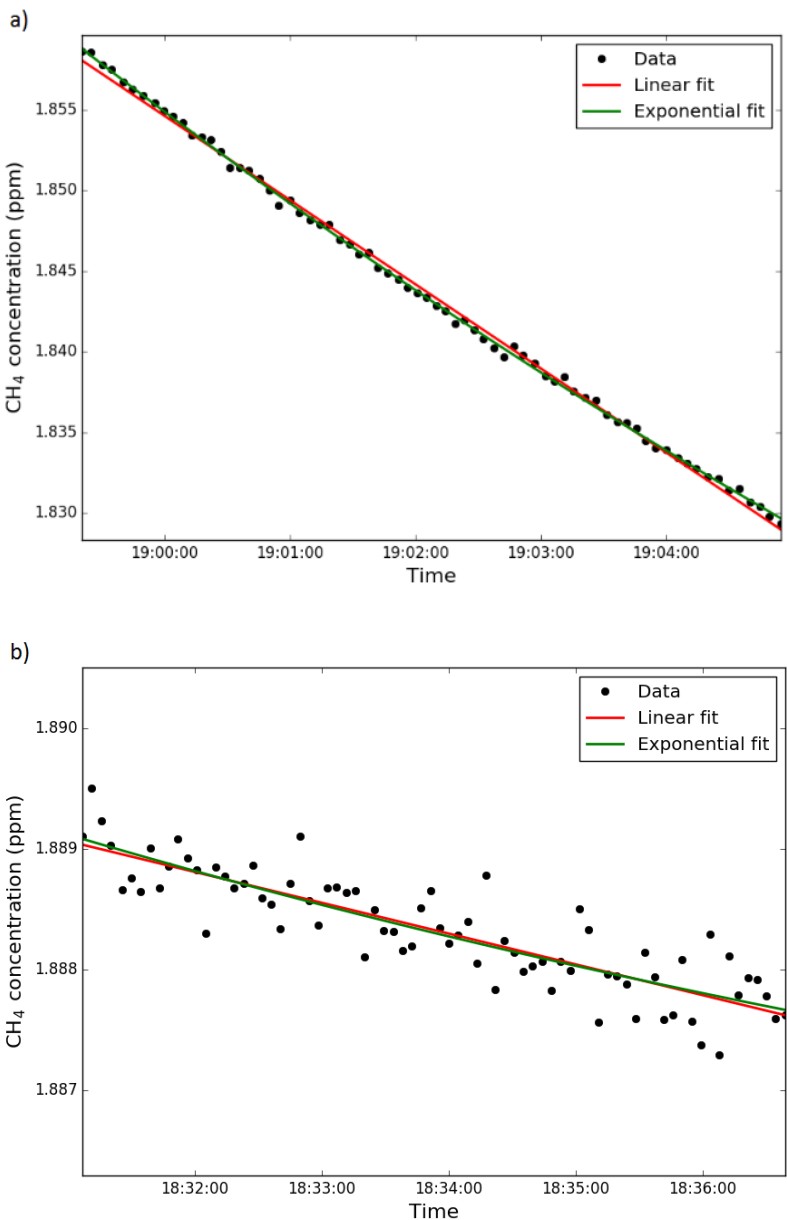

**Figure 2: Concentration data during one chamber closure for a case with a higher (a: linear and exponential: -90 and 104 µg $CH_4$ m$^{-2}$ h$^{-1}$, respectively) and lower (b: lin -3.5, exp -4.3 µg $CH_4$ m$^{-2}$ h$^{-1}$) flux.**

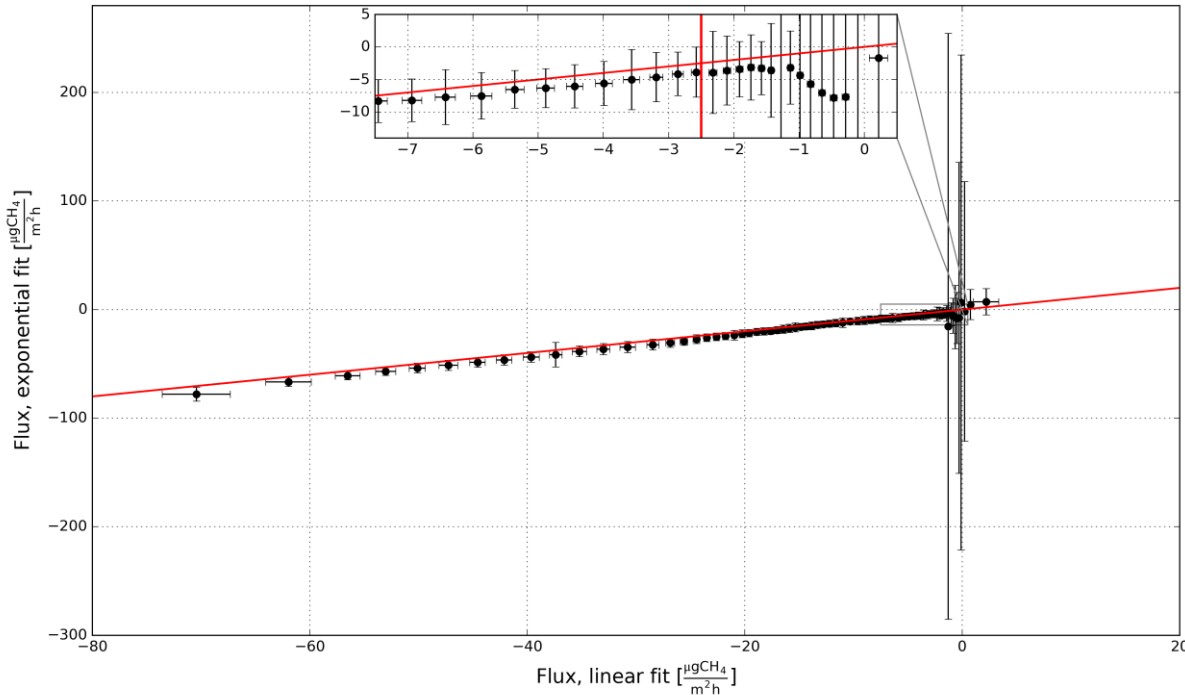

**Figure 3: Bin averages (n=500) of the linear and exponential fluxes of the whole data set (6 min closures only). In the small zoom figure the red vertical line denotes the selected flux limit of 2.5 µg CH$_4$ m$^{-2}$ h$^{-1}$. Vertical and horizontal error bars show the standard deviation of flux determined with the exponential and linear fit, respectively.**

a)

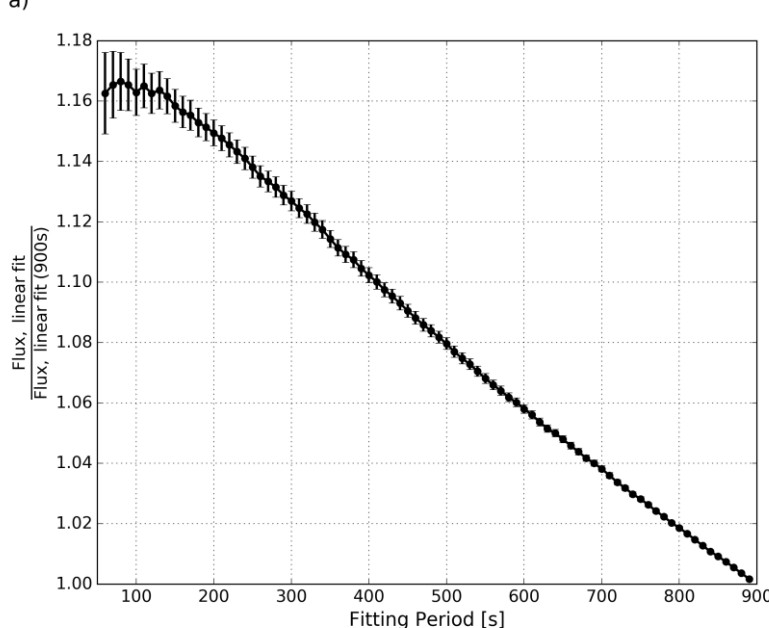

b)

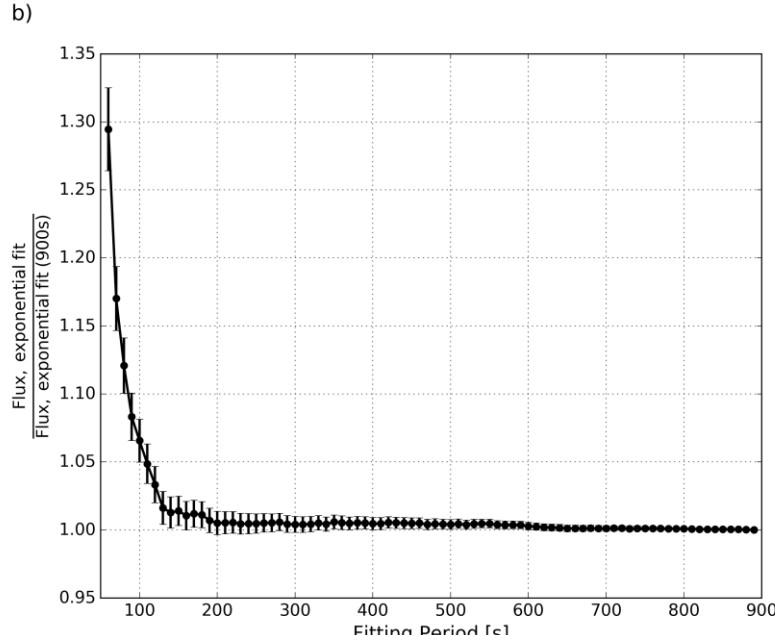

**Figure 4: The linear (a) and the exponential flux (b) as a function of fitting period. The fluxes are scaled by the flux calculated with the longest fitting period (900 s). The error bars show the 95 % confidence intervals. The data are**
5    **from summer 2012.**

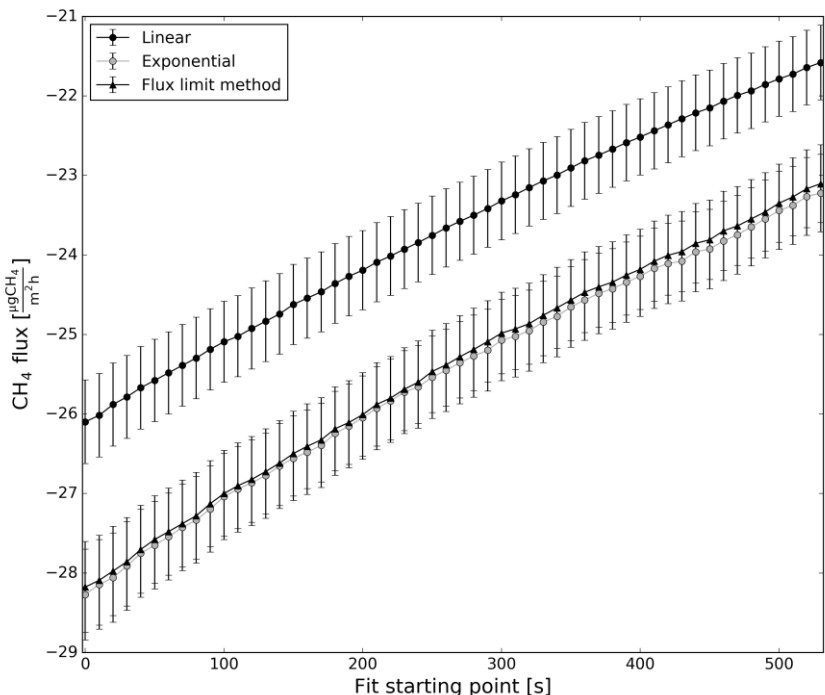

**Figure 5: Fluxes calculated with the linear, exponential and 'Flux limit' methods using a 6 min fitting period with different starting points for the fits. The error bars show the 95 % confidence intervals. The data are from summer 2012.**

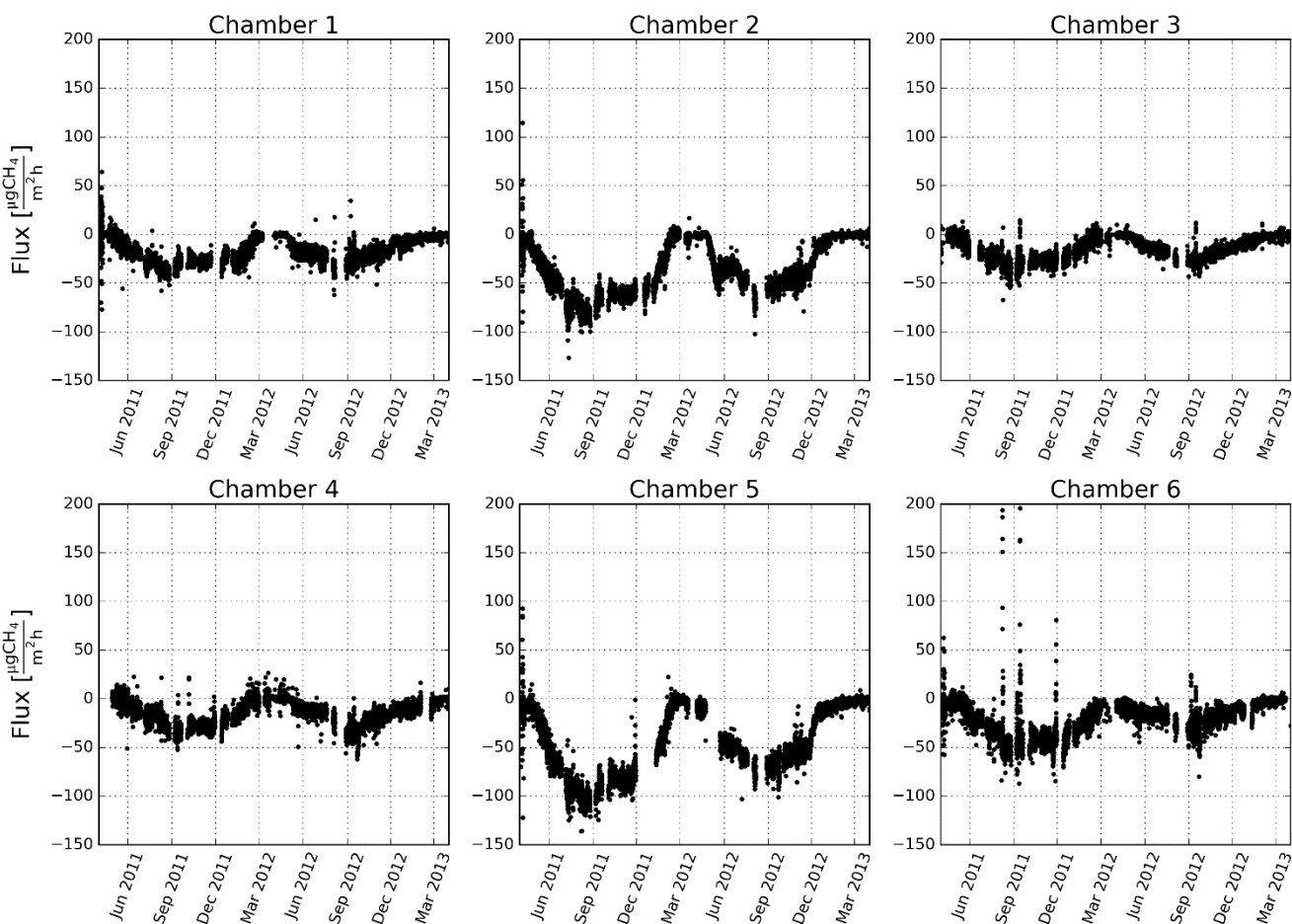

**Figure 6: Hourly CH₄ fluxes from April 2011 to March 2013 measured in each chamber. Negative values indicate uptake by the soil, and positive values indicate emission to the atmosphere. Fluxes have been calculated using the exponential fit unless the value of the flux obtained from the linear fit was below 2.5 μg CH₄ m⁻² h⁻¹.**

a)

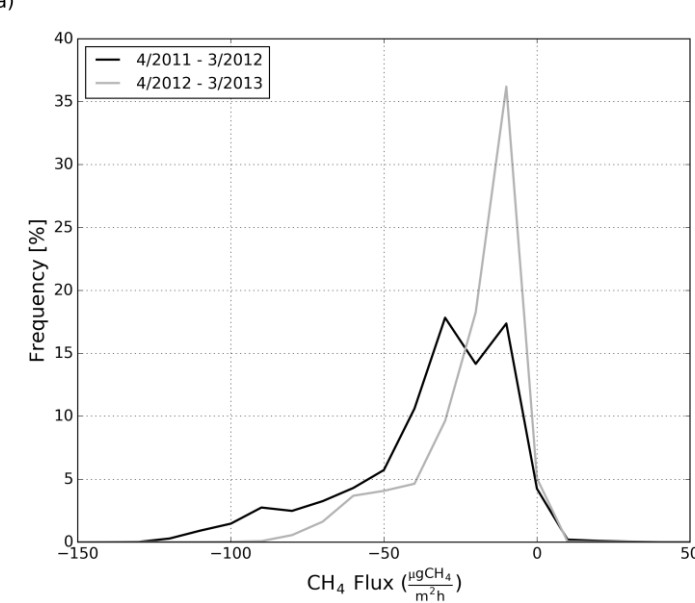

b)

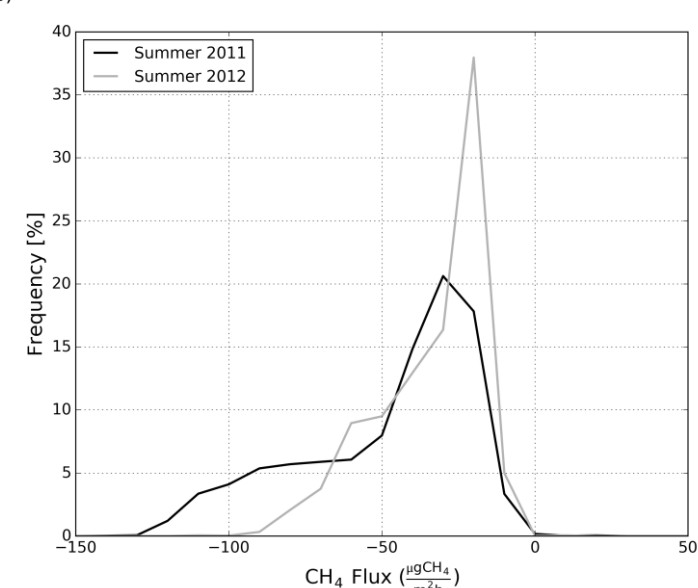

**Figure 7: The frequency distribution of fluxes measured with all chambers in (a) different years and (b) different summers. The fluxes were grouped into classes of 10 μg CH₄ m⁻² h⁻¹.**

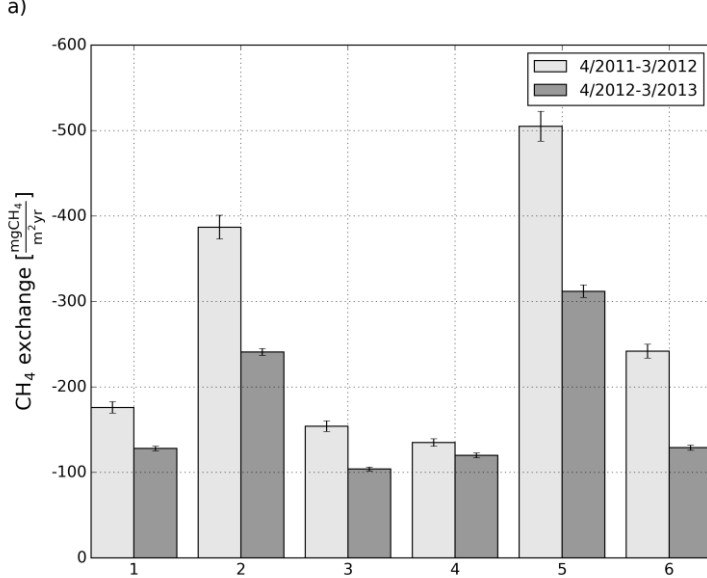

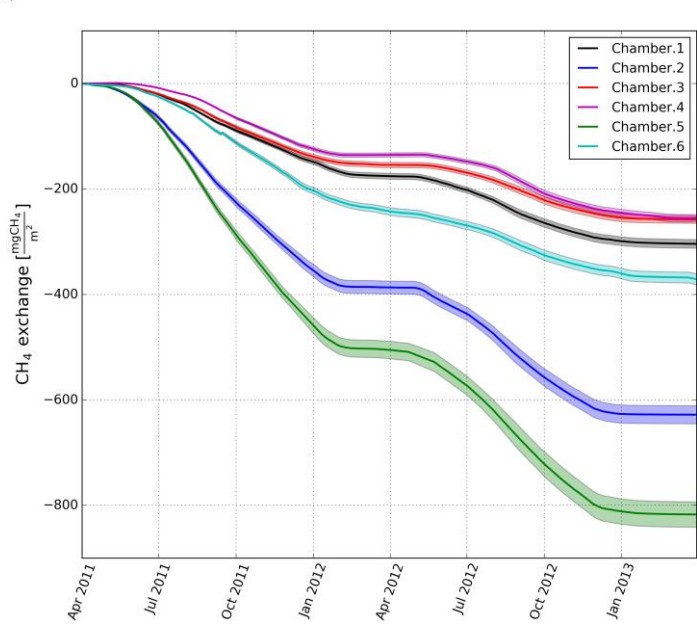

Figure 8: (a) Annual CH$_4$ exchange for each chamber for two one-year monitoring periods, and (b) cumulative CH$_4$ exchange for each chamber from 1 April 2011 to 31 March 2013. The error bars (a) and shading (b) include estimations of the random error, the error caused by gap filling and the uncertainty of the correction of the fluxes measured using the 2-min closure time (see Sect. 2.7).

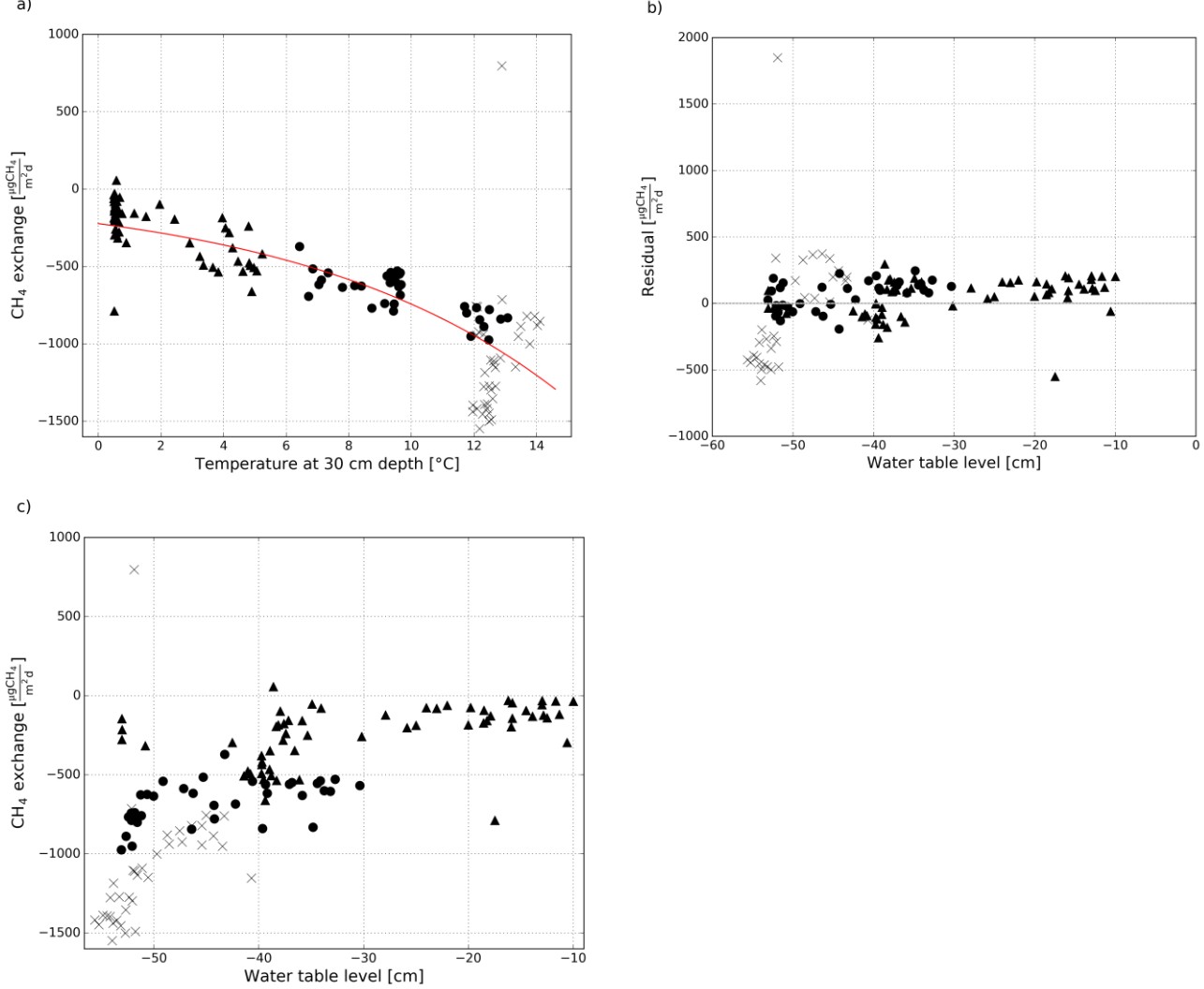

**Figure 9: (a) Daily CH₄ exchange plotted against soil temperature at 30 cm in chamber #6 for spring (April and May) (triangles), for the first half of the summer (1 June to 15 July) (circles) and for the second half of the summer (16 July to 31 August) (crosses) in 2011. Red curve in (a) denotes the exponential fitting (Eq. 2) to the data. (b) Residuals of the exponential fitting in panel (a) against water table level. (c) Daily CH₄ exchange in chamber #6 against water table level (the same flux data as in panel (a)).**

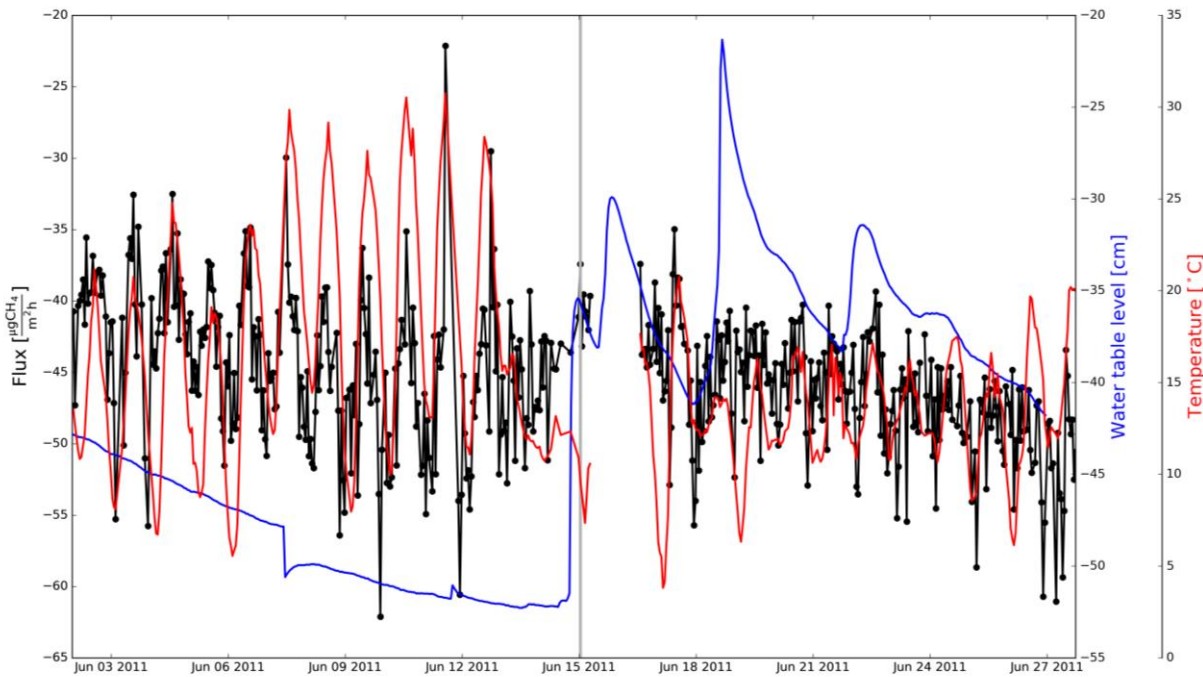

**Figure 10: Hourly CH₄ flux (black circles), air temperature (red curve) and water table level (WTL) (blue curve) in June 2011 in chamber #5. The grey vertical bar shows the split of the data for Fig 11.**

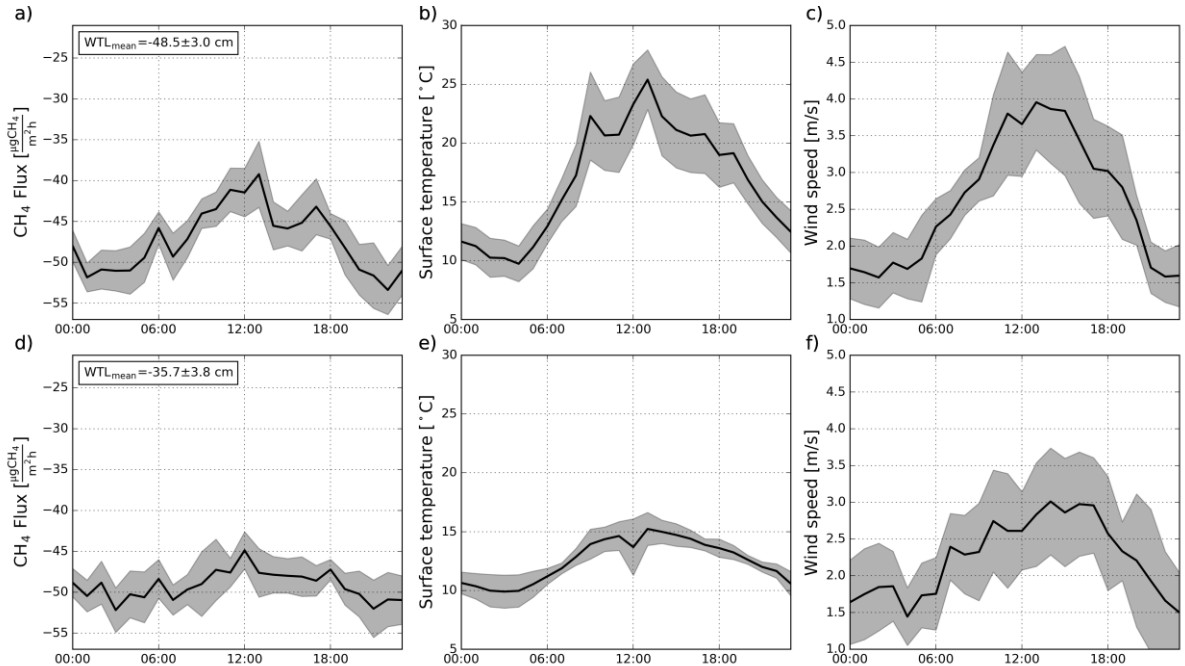

**Figure 11:** Diurnal variation of mean $CH_4$ flux (a, d) and soil surface temperature (°C) (b, e) measured in chamber #2, and the wind speed (c, f) measured above the canopy, in 2-14 June 2011 (a, b, c) and 15-27 June 2011 (d, e, f). Shading shows the 95 % confidence intervals.