# Peer review of "Methane exchange at the peatland forest floor – automatic chamber system exposes the dynamics of small fluxes"

_Biogeosciences, 2016_

## Referee Comment (RC1) · Anonymous Referee #1 · 4 Sep 2016

The authors present two years of $CH_4$ flux data from a drained peatland forest site which have been collected with an automated chamber system consisting of six chambers connected to a high resolution gas analyser. The aim of the study is twofold. First, the flux data series is used to test whether $CH_4$ fluxes from these chamber measurements are better analysed with linear or non-linear regression. As conclusion, the authors recommend to first calculate all fluxes by linear regression and then to recalculate high fluxes with an exponential regression. High $CH_4$ fluxes were defined by a site-specific threshold. Second, the study analyses the variation in $CH_4$ flux rates from the forest floor at various scales (diurnal, seasonal, inter-variation) and annual balances are presented as well.

The manuscript is very well written and this study provides an important flux dataset. High resolution gas analysers for $CH_4$ measurements are a recent development and the number of studies combining automated chambers and these analysers for long-term measurements are still scarce. This study has the potential to result in an excellent paper providing new insights into $CH_4$ flux dynamics and the methodological challenges associated with gathering these data. However in my opinion, the manuscript has two major flaws. On the one hand, the authors do not fully explore the potential of the dataset from a methodological standpoint and should expand this part of the manuscript more. On the other hand, they provide a lengthy description of the flux differences between the single chambers, but the experimental design does not really allow a proper discussion of the fluxes from an ecological standpoint. Thus, I recommend major revision and will detail my concerns below.

General comments
- I find the threshold of 3.5 µg $CH_4$ $m^{-2}$ $h^{-1}$ quite arbitrary. Based on the data presented here, I am not convinced to use such a threshold as decision for which regression to use. Why should this method be more appropriate than using a statistical criterion like e.g. AIC and to decide based on that criterion for each flux measurement separately which regression method to use?
- I am missing more details (including figures) about the effect of different closure times on the flux calculation results.
- You have only one replicate per vegetation type. Based on this setup, it is not really possible trying to understand the differences between the chambers from an ecological standpoint. Furthermore, as additional data besides vegetation composition, you only seem to have soil temperature for the single plots. Why wasn't the water table measured at each chamber? Do you have any knowledge about the soil profiles at the different locations? How has the porosity of the soil changed due to the drainage? Are your six locations really representative for the chosen vegetation types and the soil conditions at the site?
- Despite the number of replicates, the dataset is very suitable to study diurnal variations in $CH_4$ fluxes. This should be a separate section in the discussion and be more focused on the underlying processes causing these variations. Right now, this part has good references, but is mainly descriptive. In general, the manuscript has a good reference list, but often you only write which correlations were found in other studies. You need to go a step further and discuss more the processes involved. Process descriptions often stay too vague and general.

Specific comments
- Page 1, line 17: $CH_4$
- Page 2, line 14: „…thus turning in particular well-drained peatlands…"
- Page 2, line 22: Methane oxidation rates are also strongly controlled by the methane concentration in the soil, not only the oxygen concentration.
- Page 3, line 19: Add that exponential regression is especially sensitive to disturbances at the beginning of the measurement.

- Page 3, lines 19-20: It is not generally true that you need more than five data points to fit an exponential regression. It depends on the flux strength. See for example the paper by Pedersen et al. (2010) and Forbrich et al. (2010) which you cite in your manuscript. Thus, it is not uncommon to perform non-linear regression on datasets derived from syringe sampling. A high resolution gas analyser is not typically required. The great advantage of the high resolution gas analyser is that is reduces the uncertainty of the estimated slope of the flux curve, it does not necessarily change the mean estimate.
- Page 3, line 23: Specify 'high temporal resolution'. You probably mean both the sampling rate during one chamber measurement, and the total number of chamber measurements you can perform per day.
- Page 3, line 31: Do you know how much fertilizer was applied?
- Page 4, line 7: For the first species in the brackets write the full name. The way you have written it now, "S." stands for Sphagna and not Sphagnum.
- Page 4, line 20: Specify the type of fan used. What was the volume turnover inside the chamber?
- Page 4, line 31: What does "w = 1 cm" mean?
- Page 5, line 1: Is "Linak, 2009" a reference? If yes, it is not in your reference list. In general, be more consistent when mentioning product names. Include the company name as well as the associated city and country.
- Page 5, line: Wasn't the flow rate quite low? What was the actual tubing length and tubing diameter for the chambers?
- Page 5, line 7: Specify the type of sensor you used for soil temperature measurements.
- Sections 2.3 & 2.4: Which software did you use for the flux calculations and data analysis?
- Page 5, line 13: "bihourly" = twice per hour or every two hours?
- Page 5, line 14: If I understand correctly, you did not discard any data points from the measurement start. Why was it in your case justified to not apply a deadband to the flux data? How can you be sure that you had proper headspace mixing immediately after chamber closure?
- Page 5, line 17 – page 6, line 1: I don't quite understand this part. What exactly is the purpose of equation 3? Are these parameter estimates inserted into equation 4? Also, is the Kutzbach model applicable to $CH_4$ since it was developed for $CO_2$?
- Page 7, line 1: insert "$CO_2$" in front of "concentration"
- Page 7, line 10: Were these hardware problems of the gas analyser? If yes, it might be interesting information for other users.
- Pge 8, line 2: insert "it" before "usually".
- Page 8, lines 4-12: You base a lot of the following sections on these results. Provide example figures of single flux concentration curves so that the reader can judge for himself/herself.
- Page 8, line 9: How do you know that it is an underestimation?
- Page 8, line 16: Shouldn't it be "<"?
- Page 8, lines18-19: This sentence is mainly a repetition of the previous sentences. And is the data removal really the only reason for the observed differences?
- Page 8, lines 21-23: I am not convinced of this based on the presented data.
- Page 10, line 30: Friction velocity is an important parameter, but it has not been mentioned at all before this section. I assume, u* is based on the eddy tower measurements?
- Page 12, line 5: I don't see a reason for mentioning the $CO_2$ data here.
- Page 12, lines 10-11: This is an obvious observation when using relative differences.
- Page 13, line 1: Also discuss the importance of the water table depth. The low temperature does reduce metabolic activity, but methanogens are favoured by the increasing soil moisture content.

- Page 13, lines 14-23: This section is a very good example of the weakness in your study. You are lacking data on (potentially) important environmental variables and are just speculating here.
- Page 15, line 5: Could the lack of correlation be due to a lack of grass species (e.g. root exudates as food source for microorganisms) in comparison to the other study?
- Table 1: Do you have a reference for $VGA_{max}$? Include more details about the sampling method and the sampling time.
- Table 3: Did you also perform correlation tests on the entire dataset without dividing it into seasons?
- Figure 1: At what height was air temperature measured? What were the standard errors of the average water table depths? How far were the chambers away from the WTL measurement points? Maybe provide a map of the experimental site setup as a supplement.
- Figure 2: Did you also check the relationship for each year separately?
- Figure 3: It looks a bit like $CH_4$ uptake sometimes was even higher than -40 µg $CH_4$ $m^2$ $h^{-1}$ and the fluxes just went off scale. Also, what was the uncertainty of the single fluxes on average?
- Figure 5a: This bar chart is quite meaningless without some indication of the uncertainty for each bar. Do you also have a cumulative error estimate for Figure 5b?
- Figure 6: Is the daily flux just upscaled from the average hourly flux or does it represent the cumulative hourly fluxes per day? Also, I find it really difficult to distinguish between the black and blue points. It would be nice to have these plots for the other five chambers as supplement.

---

## Referee Comment (RC2) · M. Mastepanov (Referee) · 13 Oct 2016

The manuscript is based on the continuous two years $CH_4$ flux dataset obtained by automatic chamber measurement at a forestry-drained fen. The use of high-resolution gas analyzer let the authors to document low and variable fluxes, and continuity of the measurements together with their high temporal resolution allowed to estimate longtime net $CH_4$ exchange and judge the variability of the fluxes. The site chosen for the study is not an easy one for $CH_4$ flux measurements: the net result of both methanogenic and methanotrophic activities can change from small negative to small positive flux, being dependent on a combination of different factors. I highly appreciate the value of the obtained dataset, and would really like to see this study eventually published.

Unfortunately, within a number of comments following below, at least one seems to be critical and enforces me to ask for a major revision (followed by an additional review):

Equation 5 (page 6) seems to be incorrect, or, at least, has been presented incorrectly. If the time derivative is expressed in ppm/h, and the resulting flux F is also per hour, why the multiplier 3600 s/h is used? Then, I do not see a reason for a multiplier 273.15 in the numerator of the temperature fraction: the universal gas law with constant R in J mol-1 K-1 in the denominator implies temperature in K (273.15 + T in C) also in the denominator. Another way of expression for this type of flux equations (for example, Koskinen et al., 2014) operates with the standard molar volume of ideal gas instead of the universal gas constant (R), in this case ratios between the standard and the actual temperature and pressure are used. The equation 5 as it is stated in the manuscript is a mixture of these two correct approaches, and is mathematically incorrect. Formally, in the result of these two mistakes, the F values obtained with Eq.5 must be overestimated by six orders of magnitude. However, the reported fluxes seem to be of a realistic order of magnitude, while million times lower fluxes are absolutely undetectable by this type of measurements. Thus, I suppose the authors did not use the Eq.5 as it stated in the manuscript, but I can not exclude a chance that the actually used formula was also somewhat incorrect.

Talking about flux calculations, I would also ask the authors to describe what software and how was used; I can not imagine such amount of data was processed without some automated scripts/programs. Then, the description of these scripts/programs and their settings might be crucial to evaluate possible weaknesses of the calculations. How, for example, the moment of the chamber closure (t=0, page 5 line 25) was recognized by this program? As I understand from Koskinen et al. (2014), the chambers were controlled from a separate Linux PC, so operated according to its internal clock; the Picarro analyzer recorded the measured concentrations with the time stamp from its own clock – had those two being synchronized, and how often? Was the delay in gas lines between the chamber and the analyzer properly addressed?

More about the setup: was it the same as described by Koskinen et al. (2014), just with the Picarro analyzer connected in parallel to the Licor $CO_2$ analyzer? In the current manuscript it is stated (page 4 line 25): "The polycarbonate chamber was attached to a stainless steel frame (see description in Koskinen et al., 2014)." – not clear for the reader, is the reference about the frame, or the whole chamber, or the whole setup. Then some setup features are described, almost exactly in the same words as in the 2014 publication, but others (like valves, fan, etc.) are skipped. As the result, the description looks somewhat sleazy: for example (page 4 line 20), "A gas inlet tube

made of polyurethane (FESTO, OD = 6 mm, ID = 4 mm)" – but wasn't the outlet tube made of the same material and size? I would strongly suggest the authors to completely rethink section 2.2 – clearly state in the beginning, that the setup was described in details by Koskinen et al. (2014), repeat only key elements of that description without details, then clearly state what in the described measurement system is different from 2014 publication (here with all the details).

My next group of comments is related to the fact that the majority of the fluxes reported in the current study have a really low value (Fig.3). According to my back-of-the-envelope calculations, a net uptake of 20 µg $CH_4$ $m^{-2}$ $h^{-1}$ equals to $CH_4$ concentration change in the chamber of about 3 ppb over 2 minutes! Being amazed by the quality of the study, which made possible to justify such small fluxes, I have to stress the authors about the extra precaution with such data processing, interpretation and discussion.

For example, was the effect of water vapor dilution properly addressed? In $CO_2$ study (Koskinen et al., 2014) it was stated "The $CO_2$ concentration values were not corrected for water vapor dilution as the change in air humidity during measurement was small (data not shown)". In the current study, when $CH_4$ concentration in the chamber changes so tiny (3/1850=0.16%), even a small change of humidity inside the chamber during the measurement can strongly influence the result. Was $H_2O$ concentration in the gas sample measured by the same Picarro G1130 analyzer (page 4, line 24 – unfortunately, I was unable to find any information about this model in the Internet)? Was the wet or the dry mixing ratio used in the calculations? I think the water vapor dilution should be both addressed in the calculations, and discussed in the manuscript.

In opposite, with such small change in the chamber headspace $CH_4$ concentration, I think the discussion about "distortion of the vertical concentration gradient" between the soil and the headspace (mentioned many times throughout the whole manuscript) and the concentration feedback on the microbial oxidation rate (page 11 line 29) is virtually not applicable to the current study. Undoubtedly, both can be discussed, but with a clear note, that the change in the headspace concentration from 1850 to 1847 ppb $CH_4$ should practically not affect either gradient or methanotrophic activity.

In this context, I do not agree with the reasoning (for example, page 11 line 20) that the distortion of the vertical concentration gradient is the main reason for non-linearity of the concentration change in the closed chamber and the difference between the linear and exponential flux estimations. In my opinion much more important reasons are water vapor dilution (see above) and leakages (both through chamber construction and through the soil). The possible leakages are not discussed at all in the manuscript; even the fact that "when the wind speed increased, the uptake of $CH_4$ decreased" (page 15 line 8) does not seem suspicious for the authors. However, such fact often can be very clearly explained by small leaks in the chamber – see for example Pirk et al., 2016 (doi:10.5194/bg-13-903-2016), where the non-linearity of detected fluxes is directly related to the wind speed and the material of chamber sealing.

Another point I do not agree with, is an intransigent statement that "use of linear regression systematically underestimated $CH_4$ flux rates" (page 1 line 14, and many times later in the manuscripts). Such statement implies that one knows the "true" flux values, and compares them to the ones obtained by a linear regression. This is not the case in the current study (but is the

case for example in Pihlatie et al., 2013, where the flux was pre-set). Instead, the fluxes were estimated with two different mathematical methods (linear and exponential), and the results were somewhat different (Fig.2). Then the authors propose, that the linear estimation is more correct for low fluxes, and exponential – for high fluxes. This should be phrased as a proposal, as an assumption, supported by theoretical arguments and other studies, but still not as a statement, proven by this study.

Still having in mind very low magnitude of the fluxes, I would ask the authors to add, either in the main paper or in the supplementary material, a figure with two typical examples of concentration data during flux measurements – one with a high flux (over 3.5 µg $CH_4$ $m^{-2}$ $h^{-1}$), and one with a low flux – with lines for the linear and the exponential approximation over each. That will be a very sensible for the reader illustration of the measurement precision, signal-to-noise ratios, etc.

And the last general comment. Unfortunately, I have to mention somewhat careless formulations and citations in the introduction:

Page 2 line 10: "In peatlands, the net $CH_4$ flux between the soil and atmosphere is the sum of $CH_4$ production and oxidation (Dunfield et al., 1993)" – the word "sum" is never used in this publication; it was a great detailed study of both processes, but the authors never stated that they sum up to the flux. There are more processes – lateral transport (applicable to the current study with drainage ditches), subsurface storage – which affect the net fluxes as well.

Page 2 line 20: "…a lack of electron acceptors other than acetate and hydrogen are a precondition for the production of $CH_4$ (Segers, 1998; Kotsyurbenko et al., 2004)." Acetate and hydrogen are not electron acceptors! Hydrogen is the donor (in reaction with $CO_2$), acetate decays formally without donor-acceptor interaction. The publication by Segers is mentioning "alternative electron acceptors" a lot, but never stated that acetate and hydrogen are electron acceptors; the publication by Kotsurbenko et al. does not contain the words "electron" or "acceptor" in any form.

Page 2, lines 22-23: "The rate is mainly controlled by oxygen concentration, temperature and soil moisture (Boeckx and Van Cleemput, 1996)." The study by Boeckx and Van Cleemput was focused on experimental evaluation of three factors affecting methane oxidation: soil moisture, soil temperature and nitrogen ($NH_4^+$, $NO_3^-$) addition. Neither oxygen concentration nor $CH_4$ concentration were examined (were set the same in all samples); their importance for the methanotrophic oxidation was supposed to be obvious because they are reagents. Boeckx and Van Cleemput have never stated which factors are "main", but mentioned soil compaction and pH as well. So this citation is also incorrect and misleading: oxygen concentration, as well as methane concentration (as stated at page 11 line 29) and methanotrophic potential (amount and oxidation capacity of bacteria) are the factors, directly influencing $CH_4$ oxidation; temperature, moisture etc. are the factors of indirect action.

Page 2, line 25: "Closed chambers are commonly used in the measurement of greenhouse gas exchange between the forest floor and the atmosphere (e.g. Livingston and Hutchinson, 1995; Christensen et al., 1995; van Huissteden et al., 2005; Alm et al., 2007; Denmead, 2008; Forbrich et al., 2009, Koskinen et al., 2014)." The publication by Livingston and Hutchinson has only one

mentioning of a forest floor (an example of study in Brazilian rain forest), but does not say how common such studies are; the publications by Christensen et al. and by van Huissteden et al. are focused on tundra and never mention "forest".

I do not clearly remember all the publications cited in the current manuscript, and do not have enough time to check every reference. The four examples above warn me that the authors are not careful enough in their citations, so I really suggest them to check meticulously every citation in the manuscript: did the publication really state or show that? It is a big work, indeed, but it had to be done much earlier in the manuscript preparation stage.

At this stage I will not go for more specific comments and technical corrections related to the current manuscript text, as I imagine the text will be strongly changed before the resubmission. Still wish the authors to continue their work and bring their study to publication in a more carefully written form.

---

## Author Comment (AC1) · 30 Nov 2016

CH$_4$ exchange at the forest floor of a forestry-drained fen: low flux rates but high temporal variation by Korkiakoski et al.
Response to reviewer #1

**The authors present two years of CH4 flux data from a drained peatland forest site which have been collected with an automated chamber system consisting of six chambers connected to a high resolution gas analyser. The aim of the study is twofold. First, the flux data series is used to test whether CH4 fluxes from these chamber measurements are better analysed with linear or non-linear regression. As conclusion, the authors recommend to first calculate all fluxes by linear regression and then to recalculate high fluxes with an exponential regression. High CH4 fluxes were defined by a site-specific threshold. Second, the study analyses the variation in CH4 flux rates from the forest floor at various scales (diurnal, seasonal, inter-variation) and annual balances are presented as well.**

**The manuscript is very well written and this study provides an important flux dataset. High resolution gas analysers for CH4 measurements are a recent development and the number of studies combining automated chambers and these analysers for long-term measurements are still scarce. This study has the potential to result in an excellent paper providing new insights into CH4 flux dynamics and the methodological challenges associated with gathering these data. However in my opinion, the manuscript has two major flaws. On the one hand, the authors do not fully explore the potential of the dataset from a methodological standpoint and should expand this part of the manuscript more. On the other hand, they provide a lengthy description of the flux differences between the single chambers, but the experimental design does not really allow a proper discussion of the fluxes from an ecological standpoint.**

**Thus, I recommend major revision and will detail my concerns below.**

**General comments**
**− I find the threshold of 3.5 μg CH4 m−2 h−1 quite arbitrary. Based on the data presented here, I am not convinced to use such a threshold as decision for which regression to use. Why should this method be more appropriate than using a statistical criterion like e.g. AIC and to decide based on that criterion for each flux measurement separately which regression method to use?**

1. To answer this point, we have to start by briefly explaining that there was a flaw in our flux calculation and the results reported in the original manuscript. We have now recalculated all fluxes, and they are approximately 3 times as high as in the previous version. In addition, we have performed a dilution correction for CH4 concentrations, which also had an impact on the fluxes, and used a longer closure time to determine the fluxes (see comments below).

   To get back to the original comment of the referee, we tested the method (AIC) suggested by him/her. However, an AIC-based selection of the regression form proved to be partly inconsistent and resulted in increased noise in the calculated fluxes. Particularly during the low-flux period (winter, spring), the flux variability was higher than when using our 'Flux limit method' (see Figure 1A.)

[Figure]

Figure 1A. Time series of CH4 flux in 2011-2013 in each of the six chambers, calculated using the 'AIC method'.

[Figure]

Figure 1B. Time series of CH4 flux in 2011-2013 in each of the six chambers, calculated using the 'Flux limit method'. Both figures 1A and 1B include only 6 minute closures, therefore there are only 4 measurements/day in 2011 – March 2012.

Therefore we decided to keep our approach. However, we revised the criterion based on which the flux limit was chosen: we calculated bin averages of the linear and exponential fluxes of the whole data set and plotted them against each other (Fig. C). From that plot we estimated the value of the linearly calculated flux after which the flux variation (=noise) in the exponential fit increased and the shape of the relationship changed. The new flux limit is 2.5 µg CH4 m-2 h-1, which is about 20-25% of the limit presented in the original manuscript. We think that using this revised limit provides a more accurate and robust estimate of very low CH4 fluxes. Below this limit the concentration variations from which the flux is derived become increasingly affected by measurement noise and the exponential fitting becomes more prone to random perturbations to individual concentration data points and does not result in realistic flux estimates anymore.

[Figure]

Figure 1C. Bin averages (n=500) of the linear and exponential fluxes of the whole data set plotted against each other. In the small zoom figure the red vertical line denotes the selected flux limit of 2.5 µg CH4 m-2 h-1. (this Figure will be included in the revised paper)

**− I am missing more details (including figures) about the effect of different closure times on the flux calculation results.**
2. We tested different fitting windows by increasing the fitting period from the beginning (changing fitting length) (Fig. 1D) and by moving the start of the fit further (constant fitting length) (Fig. 1E). In this analysis we used data from summer 2012 when measurements were done with 16 min closure times. From this analysis it is clear that the closure time (or fitting period length) has an impact on the fluxes, particularly with closure times less than about 200 s. Therefore, instead of using the 120 s closure (and fitting) time, as done in the original manuscript, we decided to use 6 min closure time, as the noise in the fluxes was significantly reduced (this can be seen by comparing Fig.

1B above to Fig. 3 in the original manuscript). A closure time of 6 min was selected because it was used for most of our data. On the other hand, for studying the inter-annual variation, we then had to correct the 2011 fluxes (which were mostly measured with 2 min closure times) to correspond to the 6-min closure time. For this, we estimated the correction factor for each day separately by using the 6-min closure measured four times a day in 2011. The correction factor was then smoothed by using a moving average with a 2-week window. It varied between 0.9 and 1.2, being mostly 1.05, meaning that the 2011 measurements were increased about 5% due to the correction.

[Figure]

Figure 1D. Flux calculated with the increasing fitting length (shown in x-axis) plotted against the flux calculated with the 900 s fitting length. For example, the first point of the graph, showing a ratio of 1.29, is based on 60 s fitting length from the beginning, the second on 70 s long fitting, and so on. Data is from summer 2012. (This Figure will be included in the revised paper)

[Figure]

Figure 1E. Fluxes calculated with linear, exponential and 'Flux limit' methods using a 3-minute window and five different starting points. The data are from summer 2012.
(This Figure will be included in the revised paper)

[Figure]

Figure 1F. Smoothed correction factor for converting 2-min fluxes to the 6-min closure time
(Will be included as a Supplement)

**− You have only one replicate per vegetation type. Based on this setup, it is not really possible trying to understand the differences between the chambers from an ecological standpoint. Furthermore, as additional data besides vegetation composition, you only seem to have soil temperature for the single plots. Why wasn't the water table measured at each chamber? Do you have any knowledge about the soil profiles at the different locations? How has the porosity of the soil changed due to the drainage? Are your six**

**locations really representative for the chosen vegetation types and the soil conditions at the site?**

3. We realize that the use of "vegetation type" has been misleading and simply wrong. Because of the limitation of having only one gas analyzer for the six chambers, the chambers had to be located rather close to each other. Thus, the chambers were within a radius of 10 m from the measurement cabin (this information is now added to the manuscript). The locations are of the same vegetation type (Vaccinium myrtillus II type, information added to the manuscript) and being this close to each other on a well-drained, even (no hummock-hollow patterns) peatland they are unlikely to have markedly different water tables, soil temperatures deeper than at the soil surface, or soil conditions. The comparison of different vegetation types is not the aim of this study and not possible with this setup.

   We did not expect that the rather close locations on the even, well-drained site would have very different fluxes, but by placing the chambers on locations with different vegetation composition we wanted to get as much between-location variation in flux dynamics as possible for the analysis of daily/seasonal dynamics. The observed differences in ground vegetation composition mainly result from irregular shading of the tree stand (mentioned in the first paragraph of Chapter 2.1. Site description), and definitely do not indicate different vegetation types.

   To clarify the text, "vegetation type" has been changed to "vegetation composition".

   Soil profile description for the site (bulk density, CN ratio) is now added to the manuscript. This key background information was accidentally left out of the manuscript. The peat has definitely compacted due to the drainage (leading to lower porosity as well).

**− Despite the number of replicates, the dataset is very suitable to study diurnal variations in CH4 fluxes. This should be a separate section in the discussion and be more focused on the underlying processes causing these variations. Right now, this part has good references, but is mainly descriptive. In general, the manuscript has a good reference list, but often you only write which correlations were found in other studies. You need to go a step further and discuss more the processes involved. Process descriptions often stay too vague and general.**

4. After all the corrections applied in the flux calculation we found that 1) the variation originally found to take place in 2012, showing higher uptake at midday, disappeared, mainly due to the dilution correction; 2) the diurnal variation found after the dilution correction, and found to take place in both 2011 and 2012, showing lower uptake in the daytime, was mainly explained by wind speed variations (Fig. 1G). On the other hand, the air or soil surface temperature showed no or only a weak correlation with the CH4 flux (data not shown but will be added to supplement). Furthermore, inspired by Pirk et al. 2016 (see comments by referee #2), we tested the connection between the curvature parameter C in the exponential equation (see Eq. 2 in the original manuscript) and the wind speed and found, contrary to Pirk et al., that with a higher wind speed the curvature typically decreased (data will be shown in supplement). Our conclusion is that the relationship of CH4 flux with wind speed may be related to chamber leaking, as shown by Pirk et al., but also to other wind-driven processes, such as changes in the

concentration gradient within the soil that before the chamber closure is controlled by the wind speed. Thus there is no need to discuss the biological processes, as our evidence shows that these do not play a significant role or cannot be detected.

Pirk N., Mastepanov M., Lund M., Crill P. R.. Christensen T. 2016. Calculations of automatic chamber flux measurements of methane and carbon dioxide using short time series of concentrations. Biogeosciences. Vol. 13(4) s. 903–912.

[Figure]

Figure 1G. Hourly CH4 fluxes plotted against the wind speed measured above the canopy in June 2012. (This Figure will be included in the revised paper, but only for the chamber #2, the data from which were used in Figure 7 and 8 in the original MS. The rest of the wind speed relation figures for all chambers and for other time periods will be included as a Supplement).

**Specific comments**
**− Page 1, line 17: CH4**
    5. Corrected.

**− Page 2, line 14: „…thus turning in particular well-drained peatlands…"**
    6. Corrected.

**− Page 2, line 22: Methane oxidation rates are also strongly controlled by the methane concentration in the soil, not only the oxygen concentration.**
    7. Corrected.

**− Page 3, line 19: Add that exponential regression is especially sensitive to disturbances at the beginning of the measurement.**
    8. Corrected.

**− Page 3, lines 19-20: It is not generally true that you need more than five data points to fit an exponential regression. It depends on the flux strength. See for example the paper**

**by Pedersen et al. (2010) and Forbrich et al. (2010) which you cite in your manuscript. Thus, it is not uncommon to perform non-linear regression on datasets derived from syringe sampling. A high resolution gas analyser is not typically required. The great advantage of the high resolution gas analyser is that is reduces the uncertainty of the estimated slope of the flux curve, it does not necessarily change the mean estimate.**

9. Corrected.

− **Page 3, line 23: Specify 'high temporal resolution'. You probably mean both the sampling rate during one chamber measurement, and the total number of chamber measurements you can perform per day.**

10. Yes we do. Corrected.

− **Page 3, line 31: Do you know how much fertilizer was applied?**

11. The fertilizer contained 10.5% of P as raw phosphate and 12.4% of K as potassium salt. We have no documentation of the dosage used but an application of ca. 400–500 kg/ha, according to contemporary forest fertilization practices (Huikari & Paavilainen, 1968), would result in approximately 40 kg of P and 50 kg of K per hectare.

   Huikari, O. and Paavilainen, E.: Metsänlannoitus ("Forest fertilization"). Kirjayhtymä, Keskusmetsälautakunta Tapion julkaisuja, 1968.

− **Page 4, line 7: For the first species in the brackets write the full name. The way you have written it now, "S." stands for Sphagna and not Sphagnum.**

12. Corrected.

− **Page 4, line 20: Specify the type of fan used. What was the volume turnover inside the chamber?**

13. We assume that the "volume turnover" refers to the rate at which the air inside chamber volume is fully changed. One could calculate this as a relationship between the chamber volume (0.097 m3=97 L) and the flow rate of the sample gas (on average 0.9 L/min). This would give a turnover time of 108 min. However, in fact the chamber air is not replaced by ambient air, since the sample air is returned back to the chamber from the analyzers.
   The fan type was added to the text. The speed of the fan (Sunon Maglev, 1.7 W, 24 V, size: 8 cm x 8 cm) was regulated by adding a resistor into the circuit (see details in Koskinen et al. 2014). This information was added to the newly written chapter 2.2 (as requested by referee 2). It must be noted here, that there was a typo in Koskinen et al. concerning the size of the fan (8 x8 cm, not 12 x 12 cm).

− **Page 4, line 31: What does "w = 1 cm" mean?**

14. The whole chapter was rewritten by the suggestion of the other referee and this part was removed (see also the previous answer #13).

− **Page 5, line 1: Is "Linak, 2009" a reference? If yes, it is not in your reference list. In general, be more consistent when mentioning product names. Include the company name as well as the associated city and country.**

15. This reference was removed while the chapter 2.2 was rewritten. We also checked all the product and company names and made them more consisted as suggested by the referee.

**− Page 5, line: Wasn't the flow rate quite low? What was the actual tubing length and tubing diameter for the chambers?**

    16. The flow rate was originally adjusted following the recommendations in the Li840 manual. For Li840, a maximum flow rate is 1L/min. Thus, we have aimed to maintain the flow rate at about 0.8-0.9 L/min. This flow rate is also useful to avoid significant regimes of under/overpressure inside the chamber, which would probably result from significantly higher suction and blowing rates of sample gas. We do not see that this flow rate would cause any problems. There was a certain lag time (typically less than 10 s) between the chamber closure and the observable analyzer response, which is accounted for during the data treatment.

    Tubing length and id/od have been reported in Chapter 2.2. The actual length and diameter do not differ from the reported ones.

**− Page 5, line 7: Specify the type of sensor you used for soil temperature measurements.**

    17. PT4T, Nokeval Oy, Nokia, Finland. This information was added to the text.

**− Sections 2.3 & 2.4: Which software did you use for the flux calculations and data analysis?**

    18. All the calculations were made with the Python programming language (Python Software Foundation, version 2.7, https://www.python.org) using libraries: NumPy (http://www.numpy.org/), SciPy (http://www.scipy.org/), Pandas (http://pandas.pydata.org/) and matplotlib (http://www.matplotlib.org). Also, most relevant methods (e.g. fitting) are now explained in more detail in the text.

**− Page 5, line 13: "bihourly" = twice per hour or every two hours?**

    19. Every two hours. This is now clarified in the text.

**− Page 5, line 14: If I understand correctly, you did not discard any data points from the measurement start. Why was it in your case justified to not apply a deadband to the flux data? How can you be sure that you had proper headspace mixing immediately after chamber closure?**

    20. We actually discard the first 4 points, which is equivalent to 18 seconds. This is now added to the text. The fan was working all the time, even when the chamber was open.

**− Page 5, line 17 – page 6, line 1: I don't quite understand this part. What exactly is the purpose of equation 3? Are these parameter estimates inserted into equation 4? Also, is the Kutzbach model applicable to CH4 since it was developed for CO2?**

    21. The parameters in equation 3 (a2,b2 and c2) are used as an initial guess for the parameters (a17, b17 and c17) in equation 4. If this is not done, then fitting the equation 4 typically fails. However, we now removed the equations 3 and 4 from the text and only mention that the initial fit is done. We do not know any reason, why the Kutzbach model would not be applicable to CH4. Our data show similar behavior (although mostly opposite in sign) and the same laws of physics apply for CH4 as for CO2.

**− Page 7, line 1: insert "CO2" in front of "concentration"**

    22. Corrected.

**− Page 7, line 10: Were these hardware problems of the gas analyser? If yes, it might be interesting information for other users.**

23. The hardware problems here mean the problems with the chambers not working correctly. This was mentioned on Page 6, lines 26-27. The gas analyzer worked fine for the whole two year measurement period. This information was added into the text.

**− Page 8, line 2: insert "it" before "usually".**

24. Corrected.

**− Page 8, lines 4-12: You base a lot of the following sections on these results. Provide example figures of single flux concentration curves so that the reader can judge for himself/herself.**

25. Two example figures, showing closures with 'high' and 'low' uptake fluxes are now added to the revised manuscript (see also answer #7 for referee 2).

**− Page 8, line 9: How do you know that it is an underestimation?**

26. Thank you for pointing this out. We actually do not know. The term "underestimation" is replaced with the word "lower" or similar when occurring in the text. See also the answer #6 for referee 2.

**− Page 8, line 16: Shouldn't it be "<"?**

27. Corrected.

**− Page 8, lines18-19: This sentence is mainly a repetition of the previous sentences. And is the data removal really the only reason for the observed differences?**

28. This section was mostly rewritten after the new data analysis. We also removed Table 2b and recalculated the data shown in Table 2a.

**− Page 8, lines 21-23: I am not convinced of this based on the presented data.**

29. See the reply 30 above

**− Page 10, line 30: Friction velocity is an important parameter, but it has not been mentioned at all before this section. I assume, u* is based on the eddy tower measurements?**

30. Yes, u* is based on the eddy tower measurements. As discussed above, wind speed (which is correlated to u*) had an important role in explaining the flux dynamics. As wind speed is already discussed using other Figures and the Table 3, we decided to remove the u* figure and replaced u* with the wind speed measured above the canopy.

**− Page 12, line 5: I don't see a reason for mentioning the CO2 data here.**

31. We decided to remove the comparison to Koskinen et al. since they did not compare same subjects.

**− Page 12, lines 10-11: This is an obvious observation when using relative differences.**

32. This chapter was rewritten after the new data treatment, and these lines are not relevant anymore.

**− Page 13, line 1: Also discuss the importance of the water table depth. The low temperature does reduce metabolic activity, but methanogens are favoured by the increasing soil moisture content.**

33. Yes, this is true. However, during winters 2011 and 2012, the water level was not high during the whole winter. Instead, it reached a local minimum in February-March. Still, the soil moisture can be high beneath the snowpack and the frozen soil surface. We added soil moisture as one possible driver of methane production.

**− Page 13, lines 14-23: This section is a very good example of the weakness in your study. You are lacking data on (potentially) important environmental variables and are just speculating here.**

34. It is true that we did not measure WTD separately beside each chamber. This cannot be changed afterwards, as in the beginning of 2016 the site was harvested and thus changed. We have installed more WTD sensors within the area, but these data cannot be used for this older CH4 exchange data owing to the harvest.

Explaining the spatial variation was not the original purpose of the study. Instead, we wanted to cover, as much as possible, the varying ground vegetation composition with the current setup to get an average estimate of the soil CH4 exchange. It would have been of course ideal to have much more ancillary measurements; however, we consider reporting the flux dynamics of a forested peatland interesting as such and worth publication. This has also been discussed in the reply 4.

We first considered removing lines 14-23. However, the sentence about the temperature is not speculation, as it is based on measured soil temperatures. Also, removing all this discussion would give an impression of not considering the possible reasons for the observed differences at all. Therefore we decided to keep these lines, but change the wording of "… we cannot confirm its role…" to "…we can only speculate its role…".

**− Page 15, line 5: Could the lack of correlation be due to a lack of grass species (e.g. root exudates as food source for microorganisms) in comparison to the other study?**

35. This lack of correlation was likely caused by the fact that the measuring points at Lettosuo do not have plant species growing roots to anoxic layers, that would be capable of directly providing root exudates as substrate to anaerobic decomposition.

**− Table 1: Do you have a reference for VGAmax? Include more details about the sampling method and the sampling time.**

36. We do not have a reference for VGAmax as it is based on unpublished data. We added a short description of the sampling method and time to the table text.

**− Table 3: Did you also perform correlation tests on the entire dataset without dividing it into seasons?**

37. Yes we did. We also calculated correlations for monthly and annual periods, which was mentioned in the text (Page 10, line 18). We decided to show only the results of the seasonal dataset because showing the monthly data would have resulted in a vast table that would not add any significant information.

**− Figure 1: At what height was air temperature measured? What were the standard errors of the average water table depths? How far were the chambers away from the WTL measurement points? Maybe provide a map of the experimental site setup as a supplement.**

38. The height was 25.5 m, which was added to the text. SE of WTL was added to Fig. 1. One of the WTL loggers was located between the chambers #1 and #2 and the rest were located about 50 m from the chambers. We have added a map of the site as a Supplement.

− **Figure 2: Did you also check the relationship for each year separately?**
39. Yes we did. They looked quite similar except for the fact that 2012 had fewer measurement points and a lower sink. This figure was removed and replaced by another one (see Figure 1C above)

− **Figure 3: It looks a bit like CH4 uptake sometimes was even higher than -40 µg CH4 m2 h-1 and the fluxes just went off scale. Also, what was the uncertainty of the single fluxes on average?**
40. Thank you for noticing the problem with the y-axis of the figure. It is now updated. The uncertainty of single fluxes mainly varied within 2-3 %; however, for very small fluxes (such as in a cold winter of 2012-2013) it was higher, about 10%. This was added to the text.

− **Figure 5a: This bar chart is quite meaningless without some indication of the uncertainty for each bar. Do you also have a cumulative error estimate for Figure 5b?**
41. An uncertainty analysis was added to the text. We estimated the uncertainty for three most important sources: 1) random error, 2) gap-filling and 3) correction of the fluxes measured using the 2-min closure time in 2011.

− **Figure 6: Is the daily flux just upscaled from the average hourly flux or does it represent the cumulative hourly fluxes per day? Also, I find it really difficult to distinguish between the black and blue points. It would be nice to have these plots for the other five chambers as supplement.**
42. The daily flux is upscaled from the average hourly flux. The colours were changed. The plots for the other five chambers are included as a Supplement.

---

## Author Comment (AC2) · 30 Nov 2016

$CH_4$ exchange at the forest floor of a forestry-drained fen: low flux rates but high temporal variation by Korkiakoski et al.

Response to reviewer #2

**The manuscript is based on the continuous two years $CH_4$ flux dataset obtained by automatic chamber measurement at a forestry-drained fen. The use of high-resolution gas analyzer let the authors to document low and variable fluxes, and continuity of the measurements together with their high temporal resolution allowed to estimate longtime net $CH_4$ exchange and judge the variability of the fluxes. The site chosen for the study is not an easy one for $CH_4$ flux measurements: the net result of both methanogenic and methanotrophic activities can change from small negative to small positive flux, being dependent on a combination of different factors. I highly appreciate the value of the obtained dataset, and would really like to see this study eventually published.**

**Unfortunately, within a number of comments following below, at least one seems to be critical and enforces me to ask for a major revision (followed by an additional review):**

**Equation 5 (page 6) seems to be incorrect, or, at least, has been presented incorrectly. If the time derivative is expressed in ppm/h, and the resulting flux F is also per hour, why the multiplier 3600 s/h is used? Then, I do not see a reason for a multiplier 273.15 in the numerator of the temperature fraction: the universal gas law with constant R in J mol-1 K-1 in the denominator implies temperature in K (273.15 + T in C) also in the denominator. Another way of expression for this type of flux equations (for example, Koskinen et al., 2014) operates with the standard molar volume of ideal gas instead of the universal gas constant (R), in this case ratios between the standard and the actual temperature and pressure are used. The equation 5 as it is stated in the manuscript is a mixture of these two correct approaches, and is mathematically incorrect. Formally, in the result of these two mistakes, the F values obtained with Eq.5 must be overestimated by six orders of magnitude. However, the reported fluxes seem to be of a realistic order of magnitude, while million times lower fluxes are absolutely undetectable by this type of measurements. Thus, I suppose the authors did not use the Eq.5 as it stated in the manuscript, but I can not exclude a chance that the actually used formula was also somewhat incorrect.**

1. Thank you for the important observation, and apologies for this crucial error in the manuscript, which we were not aware of. The error was in the temperature fraction of the equation, which was actually used in the calculation. Also, in the text, the derivative was said to be ppm/h when the actually used derivative was ppm/s. This explains why the reported fluxes were only three times lower, not six orders of magnitude. After the correction, the flux rates are now about three times higher as originally reported. This affected for example the annual balances but did not affect the dynamics of diurnal variation or correlations. Everything in the paper to which this error affected (figures, values, comparisons to other studies) have been updated accordingly.

**Talking about flux calculations, I would also ask the authors to describe what software and how was used; I can not imagine such amount of data was processed without some automated scripts/programs. Then, the description of these scripts/programs and their settings might be crucial to evaluate possible weaknesses of the calculations. How, for example, the moment of the chamber closure (t=0, page 5 line 25) was recognized by this program? As I understand from Koskinen et al. (2014), the chambers were controlled from a separate Linux PC, so operated according to its internal clock; the Picarro analyzer recorded the measured concentrations with the time stamp from its own clock – had those two being synchronized, and how often? Was the delay in gas lines between the chamber and the analyzer properly addressed?**

2.  All the calculations were made with the Python programming language (Python Software Foundation, version 2.7, https://www.python.org) using libraries: NumPy (http://www.numpy.org/), SciPy (http://www.scipy.org/), Pandas (http://pandas.pydata.org/) and matplotlib (http://www.matplotlib.org). Also, most relevant methods (e.g. fitting) are now explained in more detail in the text.

    Unfortunately the Picarro and the PC were synchronized only after September 2012. Therefore we had to manually synchronize the Picarro data afterwards. After September 2012, the Picarro analyzer was connected to and synchronized with the Linux PC, so the problem of different time stamps was avoided. Linux PC acted both as a time (NTP) client fetching time from Finnish Meteorological Institute's time server ca. once a day, and as a time server for the Picarro analyzer.

    Example of how the system works during one chamber measurement:
    (min:sec)
    00:00 chamber lid starts to close, and the valve is switched to the chamber line. Fan has been running all the time.
    00:30 chamber is fully closed
    00:50 a flag is added to data stream indicating the source of gas (e.g. 'Chamber1'). There was a certain lag time (up to 20 sec, depending on the flow rate) between the chamber closure and the observable analyzer response, and to ensure that all the data having the flag was usable for flux calculation this higher lag was used. In addition, during the post-processing four Picarro data points, representing about 20 s, were removed from the beginning.

**More about the setup: was it the same as described by Koskinen et al. (2014), just with the Picarro analyzer connected in parallel to the Licor $CO_2$ analyzer? In the current manuscript it is stated (page 4 line 25): "The polycarbonate chamber was attached to a stainless steel frame (see description in Koskinen et al., 2014)." – not clear for the reader, is the reference about the frame, or the whole chamber, or the whole setup. Then some setup features are described, almost exactly in the same words as in the 2014 publication, but others (like valves, fan, etc.) are skipped. As the result, the description looks somewhat**

**sleazy: for example (page 4 line 20), "A gas inlet tube made of polyurethane (FESTO, OD = 6 mm, ID = 4 mm)" – but wasn't the outlet tube made of the same material and size? I would strongly suggest the authors to completely rethink section 2.2 – clearly state in the beginning, that the setup was described in details by Koskinen et al. (2014), repeat only key elements of that description without details, then clearly state what in the described measurement system is different from 2014 publication (here with all the details).**

3. Thanks for this comment. It is true, that the chapter was in some parts copied from Koskinen et al., but it was not clearly stated that the system is exactly the same. Now, the whole chapter 2.2. has been rewritten with these suggestions taken into account.

**My next group of comments is related to the fact that the majority of the fluxes reported in the current study have a really low value (Fig.3). According to my back-of-the-envelope calculations, a net uptake of 20 µg $CH_4$ $m^{-2}$ $h^{-1}$ equals to $CH_4$ concentration change in the chamber of about 3 ppb over 2 minutes! Being amazed by the quality of the study, which made possible to justify such small fluxes, I have to stress the authors about the extra precaution with such data processing, interpretation and discussion.**

**For example, was the effect of water vapor dilution properly addressed? In $CO_2$ study (Koskinen et al., 2014) it was stated "The $CO_2$ concentration values were not corrected for water vapor dilution as the change in air humidity during measurement was small (data not shown)". In the current study, when $CH_4$ concentration in the chamber changes so tiny (3/1850=0.16%), even a small change of humidity inside the chamber during the measurement can strongly influence the result. Was $H_2O$ concentration in the gas sample measured by the same Picarro G1130 analyzer (page 4, line 24 – unfortunately, I was unable to find any information about this model in the Internet)? Was the wet or the dry mixing ratio used in the calculations? I think the water vapor dilution should be both addressed in the calculations, and discussed in the manuscript.**

4. Picarro G1130 gas analyzer measures also $H_2O$ concentration, so the dilution correction is possible and it is also automatically made by the analyzer. By the referee's suggestion, we are now using the dilution corrected data, which in some cases had a very small impact on fluxes, but in some cases even the sign of the flux was changed. The correction changed the annual balances a little and some correlations with CH4 flux and meteorological and soil quantities. However, its impact to the diurnal variations was in some cases very significant. For example, we cannot see diurnal variation showing higher CH4 uptake during the daytime any more as observed in June 2012 (Fig.8 in the original manuscript). The diurnal variation did not vanish, but it is now similar to the ones observed in spring and summer 2011. This phenomena came, as the referee suggested, purely from the concurrent variation in H2O concentration. So now all the diurnal variation we see, shows lower uptake during the midday. As a whole, this was a very important comment by the referee and crucial for the quality of this study and we thank him for that, and apologize for neglecting it in the original manuscript.

**In opposite, with such small change in the chamber headspace CH₄ concentration, I think the discussion about "distortion of the vertical concentration gradient" between the soil and the headspace (mentioned many times throughout the whole manuscript) and the concentration feedback on the microbial oxidation rate (page 11 line 29) is virtually not applicable to the current study. Undoubtedly, both can be discussed, but with a clear note, that the change in the headspace concentration from 1850 to 1847 ppb CH₄ should practically not affect either gradient or methanotrophic activity.**

**In this context, I do not agree with the reasoning (for example, page 11 line 20) that the distortion of the vertical concentration gradient is the main reason for non-linearity of the concentration change in the closed chamber and the difference between the linear and exponential flux estimations. In my opinion much more important reasons are water vapor dilution (see above) and leakages (both through chamber construction and through the soil). The possible leakages are not discussed at all in the manuscript; even the fact that "when the wind speed increased, the uptake of CH₄ decreased" (page 15 line 8) does not seem suspicious for the authors. However, such fact often can be very clearly explained by small leaks in the chamber – see for example Pirk et al., 2016 (doi:10.5194/bg-13-903-2016), where the non-linearity of detected fluxes is directly related to the wind speed and the material of chamber sealing.**

5. We agree with the referee in this subject. After performing the dilution correction as suggested by the referee, the closures where the H2O concentration changed significantly, became more linear. We considered and examined the wind speed impact on fluxes, and found that the observed diurnal variation was mainly explained by the wind speed (see also answer #4 for referee 1). However, as pointed out in the answer #4, the wind speed dependency is not necessarily fully attributed to leakage, but can be related to the distortion of the soil concentration gradient by wind before the chamber closure. We also studied the curvature parameter c in relation to wind speed and found that the curvature was typically smaller with higher wind speeds (see also answer #4 for referee 1). The discussion has now been modified accordingly.

**Another point I do not agree with, is an intransigent statement that "use of linear regression systematically underestimated CH₄ flux rates" (page 1 line 14, and many times later in the manuscripts). Such statement implies that one knows the "true" flux values, and compares them to the ones obtained by a linear regression. This is not the case in the current study (but is the case for example in Pihlatie et al., 2013, where the flux was pre-set). Instead, the fluxes were estimated with two different mathematical methods (linear and exponential), and the results were somewhat different (Fig.2). Then the authors propose, that the linear estimation is more correct for low fluxes, and exponential – for high fluxes. This should be phrased as a proposal, as an assumption, supported by theoretical arguments and other studies, but still not as a statement, proven by this study.**

6.  We agree with the referee that it was wrong to use the statement "underestimation" in case of linear regression and this is now corrected in the text (see also the answer #26 for referee 1). Our intention was not to prove that the linear estimation is more correct for low fluxes, but to find a method by which the high noise in the small fluxes produced by exponential fitting could be reduced. Therefore we are suggesting, that for our data a flux limit of 2.5 µg CH4 m-2 h-1 would be an appropriate limit, and that in the forthcoming studies a similar approach could be applied, but the limit need to be estimated individually for each case. By referring to comment #2 for referee1, we think that using this limit provides a more accurate and robust estimate of very low CH4 fluxes. Below this limit the concentration variations from which the flux is derived become increasingly affected by measurement noise and the exponential fitting becomes more prone to random perturbations to individual concentration data points and does not result in realistic flux estimates anymore.

**Still having in mind very low magnitude of the fluxes, I would ask the authors to add, either in the main paper or in the supplementary material, a figure with two typical examples of concentration data during flux measurements – one with a high flux (over 3.5 µg CH4 m⁻² h⁻¹), and one with a low flux – with lines for the linear and the exponential approximation over each. That will be a very sensible for the reader illustration of the measurement precision, signal-to-noise ratios, etc.**

7.  Figures of concentration data of 'high' and 'low' flux cases have been added to the revised version of the manuscript (see also answer #25 for referee 1).

[Figure]

Figure 2A. Concentration data during one chamber closure for a case with a higher (left: linear and exponential: -90 and 104 µg CH4 m-2 h-1, respectively) and lower (right: lin -3.5, exp -4.3 µg CH4 m-2 h-1) flux.

**And the last general comment. Unfortunately, I have to mention somewhat careless formulations and citations in the introduction:**

**Page 2 line 10: "In peatlands, the net CH4 flux between the soil and atmosphere is the sum of CH4 production and oxidation (Dunfield et al., 1993)" – the word "sum" is never used in this publication; it was a great detailed study of both processes, but the authors never stated that they sum up to the flux. There are more processes – lateral transport (applicable to the**

current study with drainage ditches), subsurface storage – which affect the net fluxes as well.

Page 2 line 20: "…a lack of electron acceptors other than acetate and hydrogen are a precondition for the production of $CH_4$ (Segers, 1998; Kotsyurbenko et al., 2004)." Acetate and hydrogen are not electron acceptors! Hydrogen is the donor (in reaction with $CO_2$), acetate decays formally without donor-acceptor interaction. The publication by Segers is mentioning "alternative electron acceptors" a lot, but never stated that acetate and hydrogen are electron acceptors; the publication by Kotsurbenko et al. does not contain the words "electron" or "acceptor" in any form.

Page 2, lines 22-23: "The rate is mainly controlled by oxygen concentration, temperature and soil moisture (Boeckx and Van Cleemput, 1996)." The study by Boeckx and Van Cleemput was focused on experimental evaluation of three factors affecting methane oxidation: soil moisture, soil temperature and nitrogen ($NH_4^+$, $NO_3^-$) addition. Neither oxygen concentration nor $CH_4$ concentration were examined (were set the same in all samples); their importance for the methanotrophic oxidation was supposed to be obvious because they are reagents. Boeckx and Van Cleemput have never stated which factors are "main", but mentioned soil compaction and pH as well. So this citation is also incorrect and misleading: oxygen concentration, as well as methane concentration (as stated at page 11 line 29) and methanotrophic potential (amount and oxidation capacity of bacteria) are the factors, directly influencing $CH_4$ oxidation; temperature, moisture etc. are the factors of indirect action.

Page 2, line 25: "Closed chambers are commonly used in the measurement of greenhouse gas exchange between the forest floor and the atmosphere (e.g. Livingston and Hutchinson, 1995; Christensen et al., 1995; van Huissteden et al., 2005; Alm et al., 2007; Denmead, 2008; Forbrich et al., 2009, Koskinen et al., 2014)." The publication by Livingston and Hutchinson has only one mentioning of a forest floor (an example of study in Brazilian rain forest), but does not say how common such studies are; the publications by Christensen et al. and by van Huissteden et al. are focused on tundra and never mention "forest".

I do not clearly remember all the publications cited in the current manuscript, and do not have enough time to check every reference. The four examples above warn me that the authors are not careful enough in their citations, so I really suggest them to check meticulously every citation in the manuscript: did the publication really state or show that? It is a big work, indeed, but it had to be done much earlier in the manuscript preparation stage.

8. We apologize for our carelessness. We went through the citations and either corrected the text or removed the references.

At this stage I will not go for more specific comments and technical corrections related to the current manuscript text, as I imagine the text will be strongly changed before the

**resubmission. Still wish the authors to continue their work and bring their study to publication in a more carefully written form.**

9. As a general comment we want to notice that due to the recalculations and other changes and corrections in the manuscript the discussion part was in most parts rewritten. For example, the comparison to other studies will be changed, since the fluxes in the new version will be three times as high as previously. We also included two additional coauthors in the paper.

---

## Author Response (AR1)

Dead associate editor,

We have now carefully studied the comments from the reviewers and revised the manuscript accordingly. Some critical corrections were applied to the data analysis, which caused major changes to the results. Therefore, in addition to the comments given by the reviewers, we have made some significant changes to the text. We have also changed the title of the manuscript and added two new co-authors.

We thank you for the opportunity to revise our manuscript and hope that you and the reviewers find the changes satisfactory.

Response to reviewer #1

The authors present two years of CH4 flux data from a drained peatland forest site which have been collected with an automated chamber system consisting of six chambers connected to a high resolution gas analyser. The aim of the study is twofold. First, the flux data series is used to test whether CH4 fluxes from these chamber measurements are better analysed with linear or non-linear regression. As conclusion, the authors recommend to first calculate all fluxes by linear regression and then to recalculate high fluxes with an exponential regression. High CH4 fluxes were defined by a site-specific threshold. Second, the study analyses the variation in CH4 flux rates from the forest floor at various scales (diurnal, seasonal, inter-variation) and annual balances are presented as well.

The manuscript is very well written and this study provides an important flux dataset. High resolution gas analysers for CH4 measurements are a recent development and the number of studies combining automated chambers and these analysers for long-term measurements are still scarce. This study has the potential to result in an excellent paper providing new insights into CH4 flux dynamics and the methodological challenges associated with gathering these data. However in my opinion, the manuscript has two major flaws. On the one hand, the authors do not fully explore the potential of the dataset from a methodological standpoint and should expand this part of the manuscript more.

On the other hand, they provide a lengthy description of the flux differences between the single chambers, but the experimental design does not really allow a proper discussion of the fluxes from an ecological standpoint.

Thus, I recommend major revision and will detail my concerns below.

**General comments**

- I find the threshold of 3.5 µg CH4 m–2 h–1 quite arbitrary. Based on the data presented here, I am not convinced to use such a threshold as decision for which regression to use. Why should this method be more appropriate than using a statistical criterion like e.g. AIC and to decide based on that criterion for each flux measurement separately which regression method to use?

1. To answer this point, we have to start by briefly explaining that there was a flaw in our flux calculation and the results reported in the original manuscript. We have now recalculated all fluxes, and they are approximately 3 times as high as in the previous version. In addition, we have performed a dilution correction for CH4 concentrations, which also had an impact on the fluxes, and used a longer closure time to determine the fluxes (see comments below).

To get back to the original comment of the referee, we tested the method (AIC) suggested by him/her. However, an AIC-based selection of the regression form proved to be partly inconsistent and resulted in increased noise in the calculated fluxes. Particularly during the low-flux period (winter, spring), the flux variability was higher than when using our 'Flux limit method' (see Figure 1A.)

---

## Author Response (AR2)

Dear associate editor,

Thank you for your feedback. See below our responses to the comments given by the referees. In addition to the referee suggestions, we also made some small technical corrections to the text and updated some figures in the supplement.

Response to referee #1

**General comments:**
**- You mention a lot of table and figure numbers in the discussion. This is not necessary and the discussion should not appear as a repetition of the results section.**
    1.  Most of the references to tables and figures were removed from the discussion section.

**- You should check the reference formatting. There are some inconsistencies between references.**
    2.  All the references were checked and inconsistencies were corrected.

**Specific comments:**
**- Page 2, line 20: Do you also have a reference for the subsurface storage?**
    3.  A reference has now been added.
**- Page 11, line 11: Delete "and".**
    4.  Corrected.
**- Page 16, line 28-29: The numbers in the brackets should have a negative sign to indicate the uptake.**
    5.  Corrected.
**- Page 19, line 29: Since you used mg in the abstract, you should also use mg instead of g here.**
    6.  Corrected.
**- Figures 12 and 13: I don't think these figures are necessary. The most important information derived from these figures can be easily described in the text with a few sentences.**
    7.  We agree with the referee and removed the figures from the manuscript. These figures can still be found from the supplement.

Response to referee #2

**Current version of the manuscript is much stronger than the previous one. I am glad to see that the authors took seriously and positively all the reviewers' comments and did a big job to address them in the revised manuscript.**

**From the scientific point of view, one thing still puzzles me: CH4 flux (net uptake) gets weaker with increasing of the ambient wind speed (Fig.12), but at the same time gets more linear (Fig.13). Is it a fact, or artifact of the measurement method?**

8. We think that it is not an artefact. It could be explained as follows: in daytime with higher winds before the chamber closure the CH4 concentration in the soil surface is closer to the atmospheric concentration than in night-time when the concentration gradient in the soil gets steeper. Therefore, in daytime the microbes have more substrate (i.e. CH4) available, and during the rather short chamber closure time the decreasing CH4 concentration does not yet affect the oxidation rate, because the initial concentration was high enough for them.

**In the explanation (pages 17-18), the authors come to a discussion of artificial effect of the fan, which is opposite to the ambient wind effect: higher wind generated by a fan causes stronger CH4 uptake (page 18 line 21). If during low-wind conditions "mixing is enhanced after the chamber closure, resulting in a higher CH4 uptake in the chamber" (page 18 line 12), should one consider the high uptake at low wind to be artificial (due to the fan) overestimation of "true" natural fluxes? And the reported annual sinks – to be also somewhat overestimated? Looks like the authors lead a reader to this conclusion, but do not state it.**

9. We actually cannot know whether the annual estimate is an overestimate or underestimate – to judge this we should know the percentage of situations in which the fan speed exceeds the ambient wind speed, and vice versa. However, by employing continuous measurements we at least have caught the whole range of different diurnal conditions and thus reduce the systematic bias which

might result from making the chamber measurement only in daytime conditions, as is typically done with manual chamber sampling.

We modified the last paragraph of Chapter 4.2 as follows (new text in red):

A wind-induced diurnal cycle suggests that the current chamber set-up potentially leads to an over- or underestimate of the actual uptake rate during lower or higher fan-induced mixing, respectively, as compared to ambient mixing by wind. The chamber construction could be improved by making the fan speed vary as a function of the ambient wind speed, so as to mimic the variations in atmospheric mixing. However, we can expect that the systematic bias resulting from the wind response is minimized when employing automated sampling that facilitates continuous measurements. Our results imply that sporadic sampling with manual chambers, which is typically limited to the daytime, would have resulted in lower uptake estimates for this site than the extensive data collected with our automatic system.

**Nevertheless, I agree with the main recap of this discussion: while the fan-generated wind in a closed chamber differs from the ambient wind, it can be a source of artifacts. With this, I can approve the publication in its current form.**

[revised manuscript text omitted]